# Pathogenesis-adaptive polydopamine nanosystem for sequential therapy of ischemic stroke

Di Wu [1,3] ✉, Jing Zhou[1,3], Yanrong Zheng[1,3], Yuyi Zheng[1], Qi Zhang[1], Zhuchen Zhou[1], Xiaojie Chen[1], Qi Chen [1], Yeping Ruan[1], Yi Wang [1,2] & Zhong Chen [1] ✉

Ischemic stroke is lethal cerebrovascular disease, and reperfusion as the main strategy of blood supply restoration can cause severe ischemic brain damage. Considered as the major obstacle in medication for stroke, neuroinflammation after reperfusion undergoes dynamic progression, making precision treatment for stroke a Herculean task. In this work, we report a pathogenesis-adaptive polydopamine nanosystem for sequential therapy of ischemic stroke. Intrinsic free radical scavenging and tailored mesostructure of the nanosystem can attenuate oxidative stress at the initial stage. Upon microglial over-activation at the later stage, minocycline-loaded nanosystem can timely reverse the pro-inflammatory transition in response to activated matrix metalloproteinase-2, providing on-demand regulation. Further in vivo stroke study demonstrates a higher survival rate and improved brain recovery of the sequential strategy, compared with mono-therapy and combined therapy. Complemented with satisfactory biosafety results, this adaptive nanosystem for sequential and on-demand regulation of post-stroke neuroinflammation is a promising approach to ischemic stroke therapy.

Ischemic stroke is a lethal cerebral disease threatening public health and reperfusion including clot lysis and mechanical removal of thrombus is the most effective approach to restore blood supply[1,2]. However, more than 60% of survivors are still disabled even after complete therapy, mainly due to poststroke neuroinflammatory cascades (e.g., neuronal apoptosis, hemorrhagic transformation and etc.)[3]. Although medication of anti-inflammatory agents could reduce acute inflammatory injury, there are some disadvantages of current therapeutic strategies such as low therapeutic index, failure of neural repair and ineffective brain remodeling in the recovery period[4]. How to reduce expansion of ischemic penumbra, as well as promote brain recovery is therefore a key contributor to an improved long-term outcome. Neuroinflammation is a complex but critical pathological process after ischemic stroke, involving a variety of cell types and biochemical pathways[5]. It is well known that neuroinflammation is spatiotemporal specific to ischemic brain injury, as it plays different roles in different stages after reperfusion[6]. At the initial stage, reperfusion leads to a sharp increase of blood oxygen in the infarct area, releasing cytotoxic reactive oxygen species (ROS) and pro-inflammatory factors. Afterwards, the immune system responses in brain, and glia cells (microglia in particular) are reactive to the pathological changes, contributing to the clearance of necrotic cell debris, neural circuit repair, and synaptic reconstruction. However, under multiple effects of inflammatory factors, the microglia will be overactivated and display multivariate state changes, secreting inflammatory cytokines including tumor necrosis factor-α (TNF-α),

[1]Key Laboratory of Neuropharmacology and Translational Medicine of Zhejiang Province, School of Pharmaceutical Sciences, Zhejiang Chinese Medical University, Hangzhou 310053, China. [2]Zhejiang Rehabilitation Medical Center Department, The Third Affiliated Hospital of Zhejiang Chinese Medical University, Hangzhou 310053, China. [3]These authors contributed equally: Di Wu, Jing Zhou, Yanrong Zheng. ✉e-mail: wudichem@zju.edu.cn; chenzhong@zju.edu.cn

activated matrix metalloproteinase (MMP) and so forth, which aggravate the brain injury. Although mono-therapy at a specific stage has shown efficacy over ischemic stroke, individual difference of the pathogenesis and the short therapeutic window largely compromised the therapy in long-term recovery[7]. Therefore, we hypothesize sequential treatment of ROS scavenging and on-demand glial polarization as a pathogenesis-adaptive strategy is a potential method to address the issue of reperfusion-induced neuroinflammation.

In the past decades, nanomedicine has emerged as a powerful toolkit for enhanced therapy of ischemic stroke. Nanoparticulate antioxidants and nano-engineered drug delivery systems (DDSs) have been proved effective in stroke treatment in recent research[8–10]. For instance, inorganic and polymeric nanomaterials with high anti-oxidative activity were developed for ROS scavenging[11–13]. DDSs with brain targeting effects were also fabricated for delivery of neuroprotective agents[14,15]. However, there are at least four bottlenecks that limit their therapeutic efficacy[16–18]. Firstly, limited research has focused on sequential treatment of initial and progressive stages which are both crucial for stroke therapy. Co-delivery of antioxidants and anti-inflammatory agents results in poor efficacy because they were supposed to work at different periods. Secondly, the pathophysiological alterations after cerebral ischemia vary from patient to patient, which requires on-demand strategy for precise polarization of glia cells to regulate neuroinflammation. Recent stimuli-responsive DDSs including ROS- and pH-triggered systems remained unsatisfactory in stroke therapy. Because increased oxidative stress and low pH are pathological characteristics at the initial and the later stages of neuroinflammation, respectively. Thirdly, the presence of the blood-brain barrier (BBB) prevents most therapeutic agents including chemical drugs and biopharmaceutics from entrance to the brain. An active strategy for improved BBB permeability is highly demanded, other than the traditional method of passive penetration through damaged ischemic BBB with unpredictable distribution. Last but not the least, nanomaterial that could integrate foresaid functional motifs in one individual system is lacking.

Herein, we report a pathogenesis-adaptive DDS (T-mPDA-Pep-Mino) for sequential regulation of ROS accumulation and microglia polarization in dynamic neuroinflammation for ischemic stroke therapy. The T-mPDA-Pep-Mino nanosystem was established on mesoporous polydopamine attached by minocycline-conjugated MMP-2 responsive peptide and PEGylated brain-targeting peptide in tandem (Fig. 1a). Polydopamine is nature-inspired polymer with intrinsic capability of ROS scavenging. Tailored meso-structures on nanoparticle surface significantly increase specific surface area, providing additional reactive sites and higher drug loading capacity. Minocycline, a selective inhibitor of microglial pro-inflammatory polarization through NF-kB pathway, was conjugated with MMP-2 responsive peptide prior to use. The use of MMP-2 responsive DDSs elaborately overcomes the obstacle of administration time, avoiding aggravated injury. Meanwhile, a brain-targeted peptide that could tightly bind to low-density lipoprotein receptor-related protein-1 (LRP-1) on BBB was incorporated by using thiol-PEG polymer as a linker. Thanks to the unique chemistry of polydopamine, these functional motifs could be simply integrated within the nanosystem via Schiff's base reaction between thiols and catechols. The results showed the nanosystem treatment achieved a higher drug accumulation in the ischemic region, efficient ROS scavenging, and MMP-2 responsive polarization of microglia in vitro and in vivo, which in all contributed to sequential regulation of the pathological progress after ischemic-reperfusion injury (Fig. 1a). Our nanosystem not only offers a pathogenesis-adaptive approach to the issues that need to be addressed in different processes after ischemia stroke, but also sheds light on the development of neuroprotection and brain remodeling against other brain disorders.

## Results

### Rational design and synthesis of T-mPDA-Pep-Mino nanosystem

In this study, the T-mPDA-Pep-Mino nanosystem was fabricated on mesoporous polydopamine serving as biological glue that integrated functional motifs in one individual system[19]. At first, uniform mesoporous polydopamine nanoparticles at a size of $126.7 \pm 13.5$ nm could be prepared in a quaternary microemulsion (ethanol-water-1,3,5-trimethybenzene-Pluronic F127) as shown in Fig. 1b and c. The addition of 1, 3, 5-trimethybenzenze serving as swelling agents leads to mesopores throughout the particle. Brunauer-Emmett-Teller (BET) surface area and average pore diameter of the nanoparticles are 41.2 $m^2/g$ and 27.8 nm, respectively (Fig. 1d). The former is 193 times higher than the BET surface area of nonmesoporous polydopamine nanoparticles (0.213 $m^2/g$) at the same size (Supplementary Figure 1). Such structural improvement paves a way for efficient cargo loading and ROS scavenging. Prior to drug loading, an MMP-2 responsive peptide (Pep, Ac-CSSSGPLGIAGQSSS) with an end of thiol group was conjugated with 9-amino minocycline (Mino) via esterification activated by EDC-NHS reaction[20,21]. The sequence of -GPLGIAGQ- refers to an MMP-2 cleavable peptide, and the short sequences of -SSS- at both ends are used to improve the water solubility. Successful conjugation was confirmed by the molecular weight change of the peptide measured by matrix-assisted laser desorption/ionization time of flight mass spectrometry (Supplementary Fig. 2). Further nuclear magnetic resonance analysis also demonstrated the coupling of Mino and peptide (Supplementary Fig. 3). Measured by high-performance liquid chromatography (HPLC), drug loading capacity of the nanosystem was determined to be 104.3 mg/g, which is consistent with the thermogravimetric analysis results (Fig. 1e and Supplementary Fig. 4). To increase the delivery efficiency, a small peptide RAP-12 linked with polyethylene glycol (PEG) was anchored on the polydopamine surface. RAP-12 is a brain-targeted peptide holding high binding affinity towards LRP-1 overexpressed on BBB membrane, facilitating the drug transportation via receptor-mediated transcytosis. Analysis of molecular weight confirmed the PEGylation of RAP-12 (Supplementary Fig. 5). After grafted with the PEGylated peptide, mono-dispersity of the nanosystem was enhanced as the hydrodynamic size of the T-mPDA-Pep-Mino particles remained unchanged in consecutive seven days (Fig. 1f). In contrast, severe aggregation was observed for the unmodified drug-loaded nanoparticles (mPDA-Pep-Mino). The ζ-potential results also demonstrated successive polymer grafting of the nanosystem due to surface modification of negatively charged polydopamine (Supplementary Fig. 6). To answer the question whether the nanosystem could provide an on-demand therapeutic strategy, in vitro drug release profiles were recorded under different stimuli by HPLC. It is indicated that MMP-2 at a concentration as low as 0.1 μg/mL could trigger the drug release, of which concentration is at the same level with that in pro-inflammatory brain region (Fig. 1g)[22,23]. Higher MMP-2 concentration can hardly accelerate the release (Fig. 1h). Further in vitro investigations of drug release were conducted in the presence of other stimuli such as ROS and low pH which are also considered as pathological characteristics in ischemic stroke. Compared with the control group, negligible drug release under either stimulus was detected, indicating exclusive stimuli-responsivity of the nanosystem (Supplementary Fig. 7). To evaluate the ROS scavenging capability of the nanosystem, comprehensive study of $\cdot OH$, $\cdot O_2^-$ and $H_2O_2$ elimination was conducted (Fig. 1i and Supplementary Fig. 8). These analysis results indicated the enhanced ROS scavenging of mesoporous polydopamine in comparison with nonmesoporous nanoparticles, evidencing the structural advantages. Importantly, the functionalization of the nanosystem did not attenuate any ROS elimination nature of mesoporous polydopamine, suggesting its potential use of ROS scavenging in brain for ischemic stroke treatment.

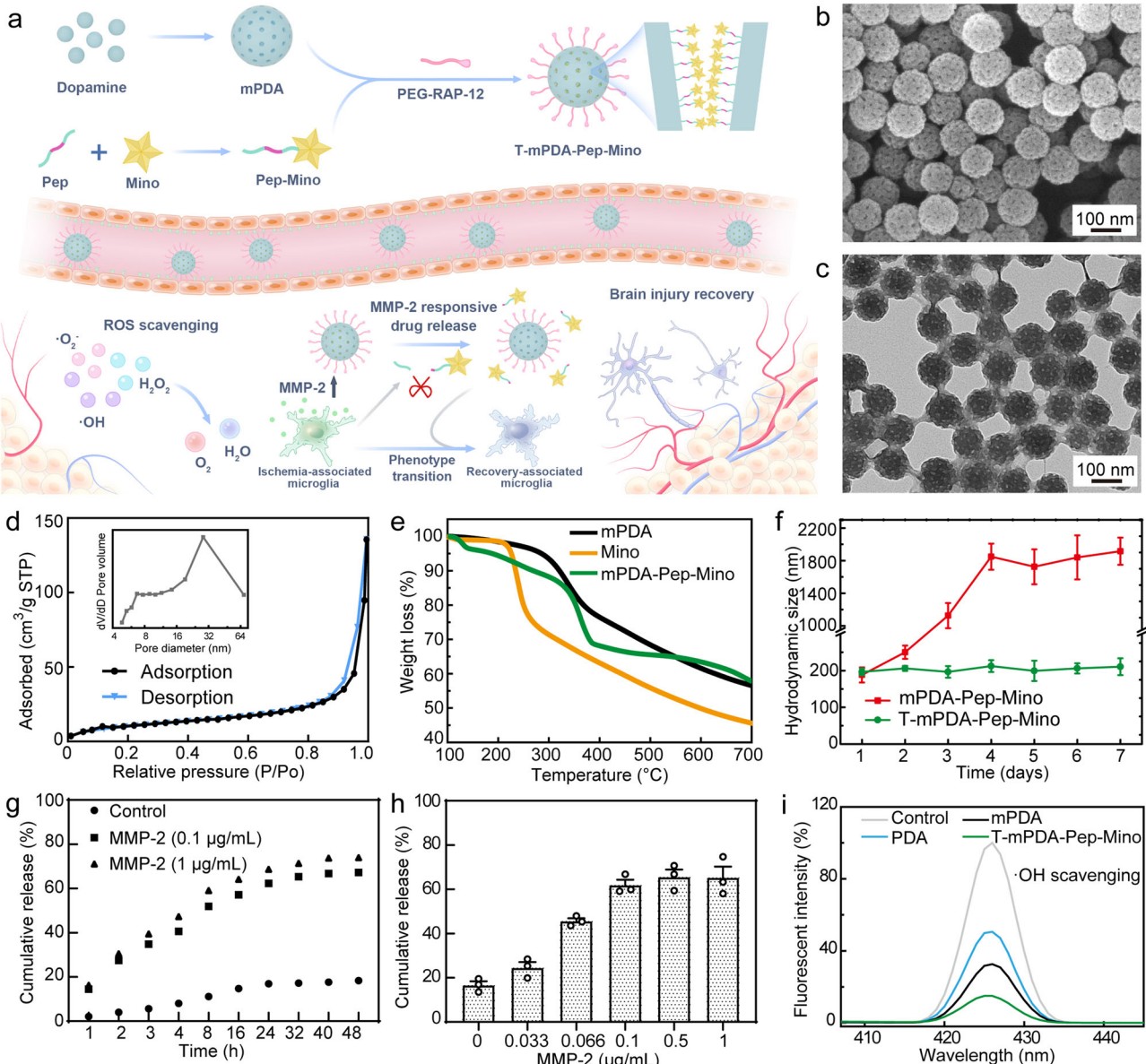

**Fig. 1 | Synthesis and characterization of polydopamine nanosystem for ischemia-reperfusion injury therapy. a** Schematic illustration of synthesis of T-mPDA-Pep-Mino nanosystem and the sequential regulation of neuroinflammation in ischemia brain. **b** Scanning electronic microscopy and **c** transmission electronic microscopy images of mesoporous nanoparticles. More than 10 times are repeated independently with similar microscopy images. **d** N$_2$ adsorption/desorption isotherms of mesoporous nanoparticles. Inset: Pore size distribution of mesoporous nanoparticles. **e** Thermogravimetry profiles of mPDA, Mino, and mPDA-Pep-Mino from 100 °C to 700 °C at a scanning speed of 10 °C per minute. **f** Hydrodynamic size changes of modified and unmodified nanoparticles in consecutive seven days in serum-containing medium ($n = 3$ independent samples). **g** Cumulative drug release profiles in 48 h in the presence of MMP-2 at 0.1 and 1 µg/mL. **h** Drug release results of the nanosystem in the presence of MMP-2 at different concentration ($n = 3$ independent experiments). **i** Evaluation of ·OH scavenging by different nanoparticles at the same particle concentration. The data are presented as means ± SEM. Source data are provided as a Source Data file.

## Neuroprotection of the nanosystem through ROS scavenging

ROS accumulation is one of the key factors of pathogenesis at the initial stage of neuroinflammation that causes neuronal injury and cell death. Thus, timely ROS scavenging by the nanosystem could be an effective approach to reduce acute ischemic reperfusion injury. To validate ROS scavenging of T-mPDA-Pep-Mino, an ROS assay kit by using 2,7-dichlorodi-hydrofluorescein diacetate as an indicator was used. As the green fluorescence of 2',7'-dichlorofluorescein indicated, both mesoporous and non-mesoporous polydopamine nanoparticles could eliminate the ROS in an SH-SY5Y cell model, in which the former featured higher efficiency at the same particle concentration (Supplementary Fig. 9). To further explore the potential effects of T-mPDA-Pep-Mino on neuroprotection, we then established an oxygen and glucose

deprivation (OGD) model of ischemic stroke in vitro. The treatment of T-mPDA-Pep-Mino up to 800 µg/mL showed merely any distinctive cytotoxicity towards SH-SY5Y cells in normal condition (Supplementary Fig. 10). As known, cellular apoptosis in ischemic stroke at the initial stage is mainly caused by oxidative stress. After 4 h of OGD followed by 24 h of re-oxygenation, significant cell apoptosis was observed via flow cytometry. Groups of both PEGylated RAP-12-modified mesoporous polydopamine (T-mPDA) and Mino exhibited neuroprotective effects. The T-mPDA-Pep-Mino nanosystem further improved the protective efficacy probably due to the combinational treatment. The apoptosis percentage of the cells treated by three groups (T-mPDA, Mino, and T-mPDA-Pep-Mino) decreased from 34.8% to 17.4%, 22.2%, and 6.8%, respectively (Supplementary Fig. 11). Intracellular ionized Ca$^{2+}$

concentration serves as another important indicator reflecting the ischemia-induced oxidative stress from mitochondria damage. As shown in Fig. 2a, the Ca²⁺ level evaluated by Fluo-8 AM staining was significantly elevated from 1.1% to 27.0% after OGD. Further treatment of T-mPDA, Mino, and T-mPDA-Pep-Mino reversed the Ca²⁺ overload to 11.0%, 16.0%, and 7.1%, respectively, suggesting a synergistic effect of neuroprotection from both the carrier and the drug. Fluorescence staining of terminal-deoxynucleotidyl transferase-mediated nick end labeling (TUNEL) for detecting DNA fragmentation in ongoing apoptosis was conducted in each group. The results in Fig. 2b clearly confirmed OGD-induced apoptosis and neuroprotective effects of T-mPDA-Pep-Mino treatment, as the distributed green fluorescence of labeled double-strand DNA breaks disappeared after T-mPDA-Pep-Mino treatment. Apoptosis is a process of programmed cell death lead by the biochemical events that were triggered by several apoptosis-regulatory genes. B-cell leukemia-2 (Bcl-2) and Bcl-2-associated X protein (Bax) are considered as two key apoptotic regulators that serving as a rheostat determining the apoptosis (Fig. 2c). As shown in Fig. 2d, the protein expression ratio of Bcl-2 to Bax decreased from 0.90 to 0.56 in OGD

group. By contrast, the treatment of T-mPDA-Pep-Mino could reduce the susceptibility of cellular apoptosis as the protein expression of Bcl-2 rose from 0.37 to 0.62 and the expression of Bax went down from 0.66 to 0.49. At the meantime, the autophagic activity of the OGD-treated cells before and after different treatments were recorded. The protein level of p62, a classic receptor of autophagy, was reduced in OGD group accompanied by an increase of LC3-II/LC3-I (Supplementary Fig. 12). By contrast, the T-mPDA-Pep-Mino group showed contrary changes of p62 protein level and ratio of LC3-II/LC3-I, suggesting its suppressive effects on autophagic activity. These results collectively indicated that the nanosystem could relieve the oxidative stress of OGD-treated cells and decrease OGD-triggered apoptosis and autophagy levels, contributing to higher cell viability.

## In vitro evaluation of MMP-2 responsive regulation of microglia polarization

Microglia are activated after ischemia injury. However, microglia are highly dynamic and plastic cells that play divergent roles in different stages at post-refusion depending on their morphology, motility,

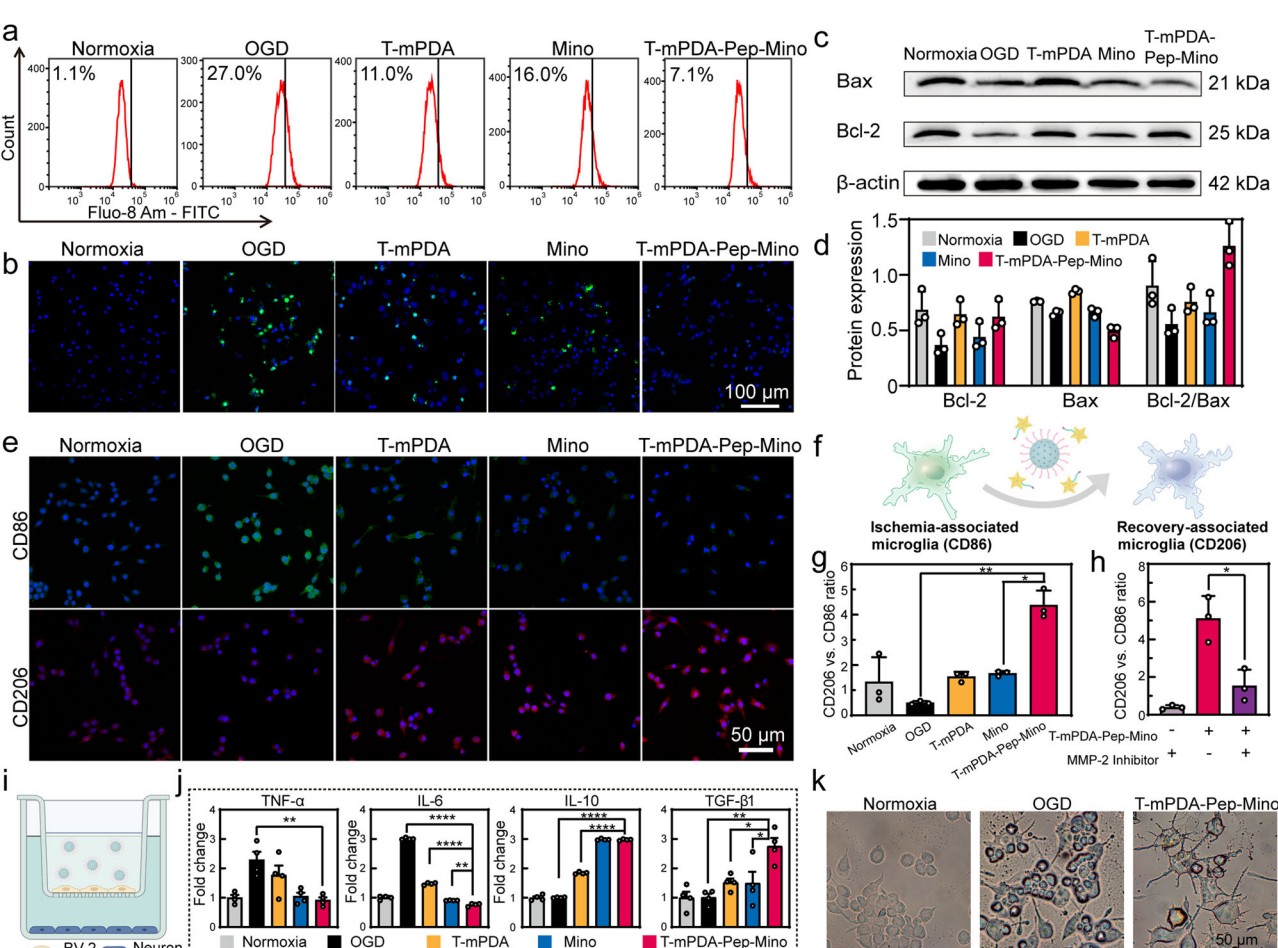

**Fig. 2 | In vitro evaluation of neuroprotection and microglia regulation of MMP-2 responsive polydopamine nanosystem for ischemia-reperfusion injury treatment.** Normoxia group was taken as the control group and the cells in other four groups were treated by OGD/re-oxygenation and corresponding solutions including saline, T-mPDA, Mino, and T-mPDA-Pep-Mino at a Mino concentration of 10 µg/mL. **a** Comparison of intracellular calcium concentration by Fluo-8 AM. **b** TUNEL-DAPI co-staining in five groups after corresponding treatment. **c** Protein expression of Bax and Bcl-2 in five groups confirmed by Western blotting and **d** corresponding analysis of expression ratio of Bcl-2 to Bax (*n* = 3 independent samples). **e** Immunostaining of CD86 and CD206 of BV-2 cells in five groups. **f** Scheme of microglia phenotype transition induced by the nanosystem. **g** Fluorescence intensity ratio changes of CD206 versus CD86 in five groups (*n* = 3).

**h** Fluorescence intensity ratio changes of CD206 versus CD86 of BV-2 cells in the absence or presence of T-mPDA-Pep-Mino or MMP-2 inhibitor (*n* = 3). Images with a cell density higher than 30 cells per image were used for evaluation. **i** Scheme of transwell study on co-culture of BV-2 cells (upper) and SH-SY5Y cells (down). Created with BioRender.com. **j** Relative expression of pro-inflammatory factors (TNF-α and IL-6) and anti-inflammatory factors (IL-10 and TGF-β1) confirmed by ELISA assay (*n* = 4). **k** Morphological observation of BV-2 cells under bright field in different conditions including Normoxia, OGD and OGD with T-mPDA-Pep-Mino treatment. More than 5 times are repeated independently with similar microscopy images. The data are presented as means ± SEM. Source data are provided as a Source Data file.

function, metabolism and etc[24]. On-demand regulation of microglia polarization in response to MMP-2 excreted by activated microglia with pro-inflammatory effects is a promising solution to avoid deterioration. To demonstrate the MMP-2 responsive regulation by the nanosystem, an OGD model was established by using BV-2 cell line. Immunofluorescence staining of pro-inflammatory marker CD86 and anti-inflammatory marker CD206 was conducted to evaluate the microglia transition after different treatment in oxidative conditions (Fig. 2e and f). Green fluorescence of CD86 marker was elevated and the ratio of CD206/CD86 was lowered in OGD group, indicating the pro-inflammatory polarization of microglia. With Mino treatment (10 µg/mL), the OGD-induced BV-2 cells were polarized into the neuroprotective phenotype. Interestingly, the treatment of T-mPDA also showed the capability of regulation of microglia polarization, which may result from its efficient ROS scavenging[25]. Compared with the OGD group, significant ratio change of CD206/CD86 in the T-mPDA-Pep-Mino group was observed (Fig. 2g) (**$P < 0.01$ compared with the control group and *$P < 0.05$ compared with the Mino group via paired $t$-test). This result could be attributed to the synergistic effect of mesoporous polydopamine and released Mino. To confirm the crucial role that MMP-2 may play in drug release, MMP-2 inhibitor was added in advance and further fluorescence imaging was conducted. Fluorescence analysis proved that the ability to promote state transition of T-mPDA-Pep-Mino nanosystem was severely inhibited in the presence of the inhibitor (Fig. 2h and Supplementary Fig. 13) (*$P < 0.05$ compared between the groups with and without inhibitor via paired $t$-test), suggesting MMP-2 is of vital importance in responsive regulation.

To give a more comprehensive characterization of the microglial response, transcriptome analysis of BV-2 cell line under different treatments was performed. Gene enrichment analysis of the cells based on Kyoto Encyclopedia of Genes and Genomes was shown in Supplementary Figs. 14–17, giving top 20 pathway enrichment of two selected groups for comparison. Compared with Normoxia group, the OGD-treated cells were involved in multiple biological processes of innate immunity, response to virus, cytokine signaling, suggesting a pro-inflammatory subtype that was termed as ischemia-associated phenotype. By contrast, in addition to the combined advantages of T-mPDA (ferroptosis) and Mino (oxidative phosphorylation, viral defense), TNF-α signal pathway and NF-κB signaling pathway were also enriched after T-mPDA-Pep-Mino incubation. Given the critical role of these pathways in pro-inflammatory cytokine production. These data implied that T-mPDA-Pep-Mino may transmit BV-2 cells from the ischemia-associated phenotype to a state with anti-inflammatory cytokine production, altered state of ferroptosis, oxidative phosphorylation, viral defense, and etc., which was termed as recovery-associated phenotype. Quantitative real-time PCR also confirmed the anti-inflammatory cytokine production under T-mPDA-Pep-Mino treatment (Supplementary Fig. 18). A heatmap of gene expression analysis indicated that T-mPDA-Pep-Mino could reverse the gene regulation by OGD treatment, whereas T-mPDA and Mino showed no significant impact on up- or down-regulation (Supplementary Fig. 19), indicating anti-inflammatory property of T-mPDA-Pep-Mino.

To validate the protective effects of microglia at recovery-associated phenotype on neurons, a transwell model with co-cultured BV-2 and SH-SY5Y cells was used to reveal the crosstalk between microglia and neuron (Fig. 2i). OGD-induced BV-2 cells on upper chamber was treated with T-mPDA-Pep-Mino after re-oxygenation. Levels of pro-inflammatory cytokines including TNF-α and IL-6 were found to be significantly decreased (Fig. 2j). (*$P < 0.05$, **$P < 0.01$, ****$P < 0.0001$ compared between the groups as indicated via one-way ANOVA with Tukey's multiple comparisons test). By contrast, anti-inflammatory cytokines including IL-10 and TGF-β1 were increased after the treatment, indicating the reduced inflammation. Further Live/Dead assay revealed that T-mPDA-Pep-Mino could improve the cell viability (Supplementary Fig. 20). Resident microglia are the major immune cells in the brain with morphological plasticity. Bright-field imaging showed representative morphologic changes of BV-2 cells after the treatment of T-mPDA-Pep-Mino (Fig. 2k). Most resting cells were polarized into de-ramified state induced by OGD treatment, indicating the microglial activation. Thanks to the selective modulation of T-mPDA-Pep-Mino, the amoeboid microglia at an activated state transformed into an elongated shape. Such morphological changes could be beneficial for microglia of recovery-associated phenotype to maintain the homeostasis of the brain. Less cell debris in the T-mPDA-Pep-Mino group also supported the conjecture. These results demonstrated that the T-mPDA-Pep-Mino nanosystem could transit microglia from ischemia-associated phenotype to recovery-associated phenotype in response to MMP-2, assisting neuroprotection after ischemia reperfusion.

## Brain protection of T-mPDA-Pep-Mino in ischemia-reperfusion injury

To verify the brain protection of T-mPDA-Pep-Mino in ischemia-reperfusion injury in vivo, a transient middle cerebral artery occlusion (MCAO) model was established according to our previous report (Fig. 3a)[26,27]. In brief, the model was developed by inserting a filament into the middle cerebral artery of the mice for 60 min, followed by removal of the filament to achieve reperfusion. Different therapeutic treatments including saline, T-mPDA, Mino, and T-mPDA-Pep-Mino at a Mino concentration of 10 mg/kg were then performed after the reperfusion. Pharmacokinetic analysis of Mino and T-mPDA-Pep-Mino was conducted by measuring the serum drug concentration at different timepoint post-injection (Supplementary Fig. 21). The concentration-time profiles revealed that the $C_{max}$ and $AUC_{0-24}$ of T-mPDA-Pep-Mino ($C_{max} = 10.0 \pm 1.3$ µg/mL; $AUC_{0-24} = 73.1 \pm 3.0$ µg/mL·h) are higher than direct administration of Mino ($C_{max} = 7.7 \pm 0.8$ µg/mL; $AUC_{0-24} = 52.0 \pm 4.4$ µg/mL·h). To evaluate the brain-targeted delivery, the nanosystem was labeled by fluorescent tags (Cyanine5.5). As demonstrated by ex vivo imaging in Fig. 3b, the fluorescence intensity of T-mPDA-Pep-Mino group is calculated to be 451% higher than that of nontargeting group (mPDA-Pep-Mino), indicating improved brain-targeted delivery by PEGylated RAP-12 peptide (Supplementary Fig. 22). The fluorescence intensity of the brain region reached the peak at 4 h post-injection and maintained at a high level within the following three days (Supplementary Fig. 23). Those unpenetrated nanoparticles may be eliminated by liver and kidney (Supplementary Fig. 24). More importantly, further imaging analysis of brain slices from each group showed higher fluorescence signals (152%) observed in the ischemic penumbra (right side). This phenomenon clearly confirmed the enhanced targeting ability of T-mPDA-Pep-Mino towards the brain lesion. After 24 h of reperfusion, the infarct volume was measured by triphenyl tetrazolium chloride (TTC) staining, and neurological deficit scores of the mice were determined by Zea-Longa score which contains motor, sensory, reflex and balance examinations[28]. Compared with Sham group, distinctive cerebral infarct was observed in brain slices of MCAO mice (Fig. 3c, d) (*$P < 0.05$, **$P < 0.01$, ****$P < 0.0001$ compared between the groups as indicated via one-way ANOVA with Tukey's multiple comparisons test). However, the treatment of T-mPDA-Pep-Mino significantly reduced the infarct volume, as well as lowered the neurological deficit scores (Fig. 3e) (*$P < 0.05$, ****$P < 0.0001$ compared between the groups as indicated via one-way ANOVA with Tukey's multiple comparisons test), suggesting its therapeutic efficacy in alleviating acute injury. As shown in Fig. 3f, the survival rate of the MCAO mice was increased within seven days after a single dose of T-mPDA-Pep-Mino (*$P < 0.05$ via survival curve comparison with Log-rank test). Behavior study at day 3 and day 7 also revealed that the treatment of T-mPDA-Pep-Mino gave lowered neurological deficit scores of MCAO mice than the treatment of either T-mPDA or Mino alone

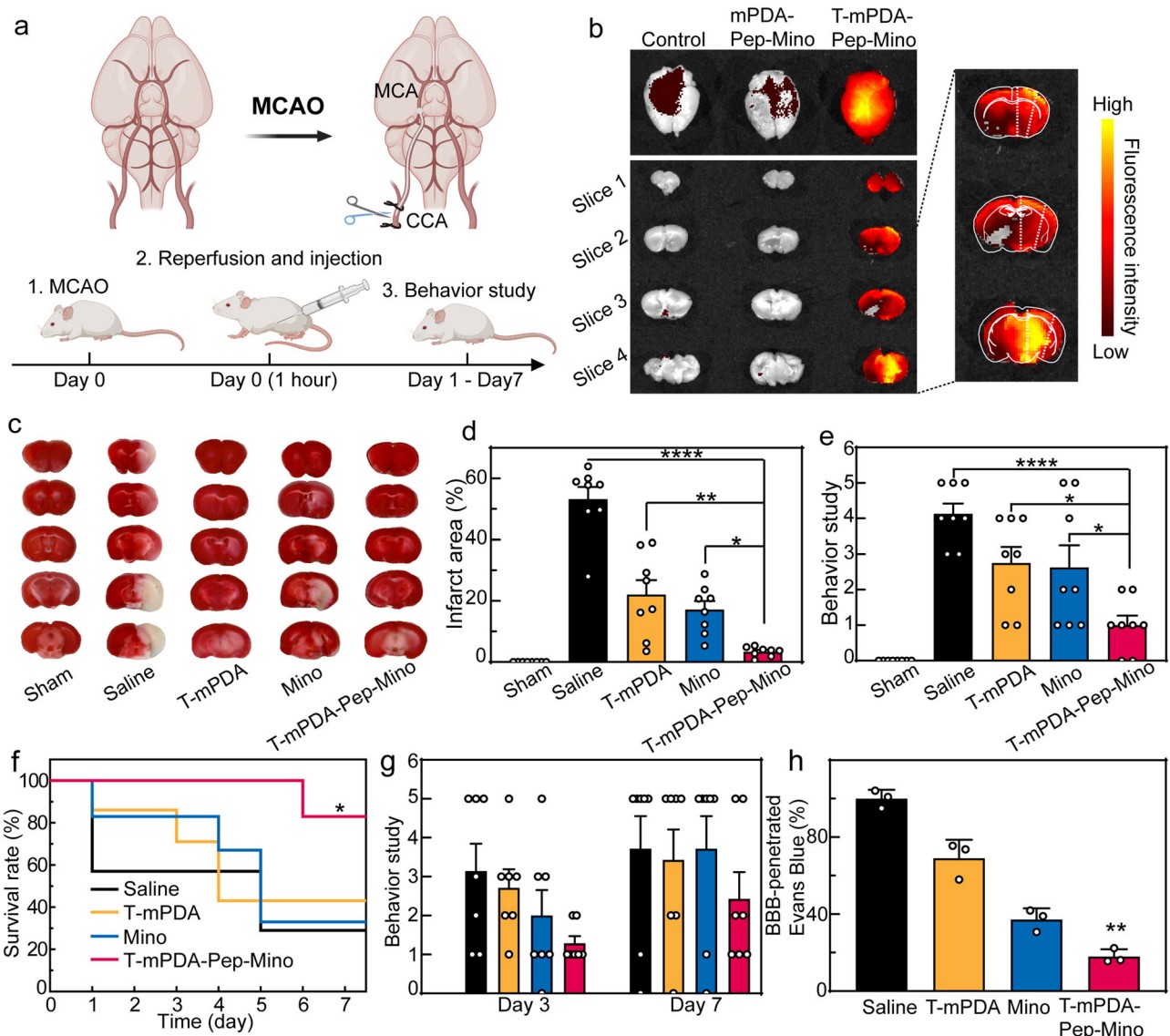

**Fig. 3 | In vivo brain protection of the nanosystem in MCAO-induced ischemia-reperfusion injury.** Sham group was taken as the control group and the mice in other four groups were treated by MCAO and corresponding administration of saline, T-mPDA, Mino, and T-mPDA-Pep-Mino at a Mino concentration of 10 mg/kg. **a** Scheme of MCAO-induced ischemia-repefusion injury model. Created with BioRender.com. **b** Ex vivo fluorescence imaging of brains and brain slices from the mice treated by free Cy5.5 (control), non-targeted nanosystem (mPDA-Pep-Mino) and brain-targeted nanosystem (T-mPDA-Pep-Mino). Magnified section: the right side of brain slices indicates the ischemic area. **c** Representative TTC staining images of brain slices in five groups. **d** Infarct volume analysis of brain slices in five groups ($n = 8$ independent animals). ****$P < 0.0001$, **$P = 0.0011$, *$P = 0.0239$, compared between the groups of T-mPDA-Pep-Mino and Saline, T-mPDA, and

Mino. **e** Neuronal function evaluation of the mice by neurological scoring ($n = 8$ independent animals). ****$P < 0.0001$, *$P = 0.0237$, *$P = 0.0412$, compared between the groups of T-mPDA-Pep-Mino and Saline, T-mPDA, and Mino. **f** Survival rate of the MCAO mice after different treatments within seven days ($n = 6$ independent animals). *$P = 0.0454$, compared between the groups of T-mPDA-Pep-Mino and Saline. **g** Neuronal function evaluation of the MCAO mice by neurological scoring at day 3 and day 7 after the injury ($n = 7$ independent animals). **h** BBB permeability evaluation of the MCAO mice by EB staining at day 7 after different treatments ($n = 3$ independent animals). **$P = 0.0014$, compared between the groups of T-mPDA-Pep-Mino and Saline. A two-tailed $P$ value of $<0.05$ was considered statistically significant. The data are presented as means ± SEM. Source data are provided as a Source Data file.

(Fig. 3g). BBB permeability of the MCAO-treated mice at day 3 post-injection was confirmed by quantitative analysis of extravasated Evans Blue (EB). The mice treated by T-mPDA-Pep-Mino had the lowest EB exudation in all four groups. (Fig. 3h) ($P^{**} < 0.01$ compared with the saline group via paired $t$ test). These results demonstrated T-mPDA-Pep-Mino treatment could protect the brain from ischemia-reperfusion injury.

**In vivo study of sequential therapy and brain recovery**
Timely treatment of ischemia-reperfusion injury at different stages after the injury is the key to enhance the therapeutic efficacy.

However, individual difference of stroke patients from one to another makes it difficult to seize the moment for precise regulation of microglia polarization. It was revealed that MMP-2 levels in the brain within 72 h post-reperfusion varied significantly during the inflammatory progression (Supplementary Fig. 25). Drug release profiles in Fig. 1h suggested that MMP-2 levels higher than 0.1 µg/mL could facilitate the on-demand regulation. To verify the performance of pathogenesis-adaptive therapy, comparative study on sequential regulation was conducted. As shown in Fig. 4a, the MCAO mice were treated by three different patterns of administration including T0-M0 (T-mPDA at day 0 and Mino at day 0), T0-M3 (T-mPDA at day 0

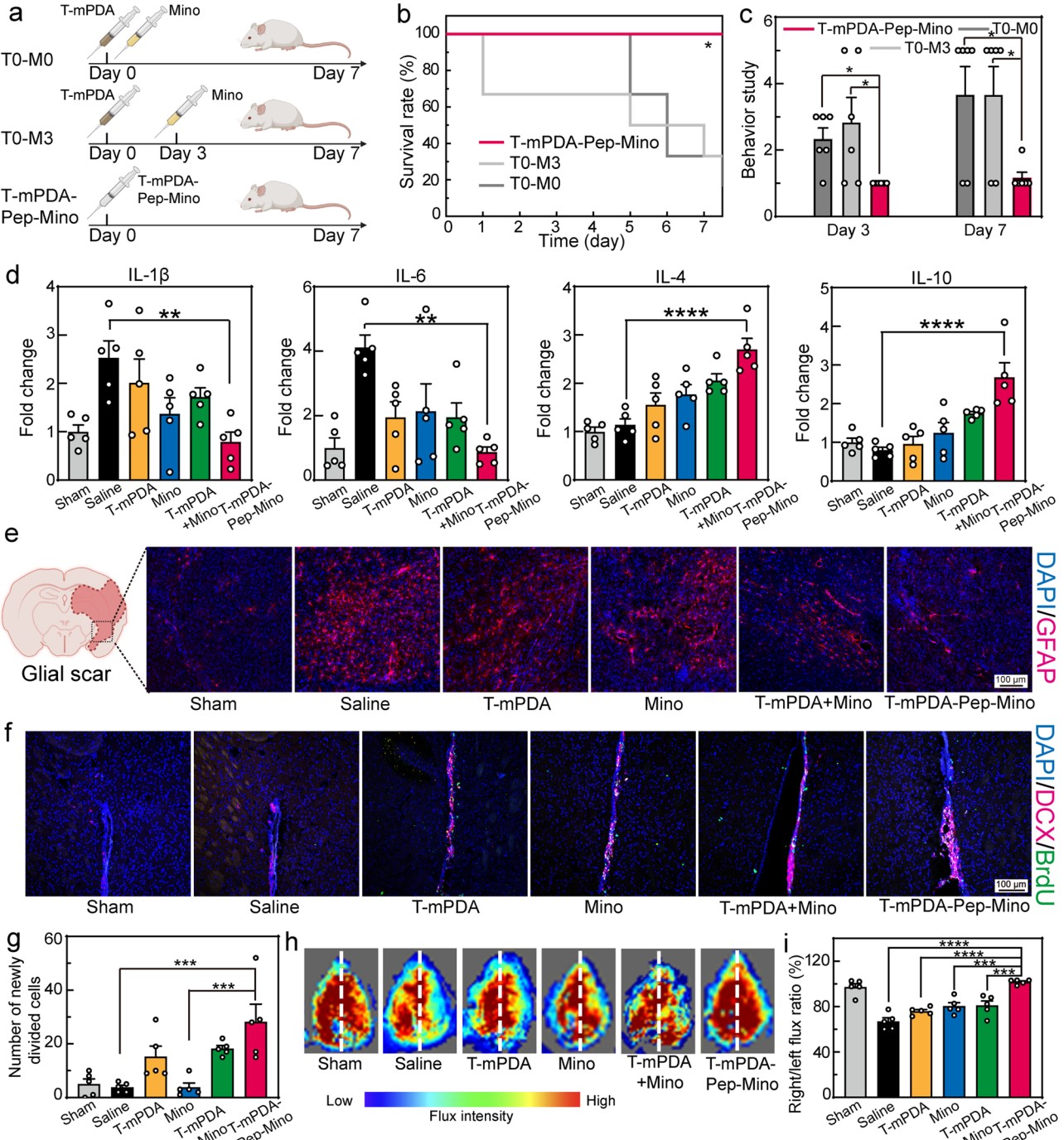

**Fig. 4 | Sequential regulation of neuroinflammation after ischemia-reperfusion injury and brain remodeling in vivo. a** Scheme of comparative study on sequential therapy in the group of T0-M0 (T-mPDA at day 0 and Mino at day 0), T0-M3 (T-mPDA at day 0 and Mino at day 3), and T-mPDA-Pep-Mino. Created with BioRender.com. **b** Survival rate of the MCAO mice after three treatments within seven days (*n* = 6 independent animals). *P = 0.0229 between the survival curves. **c** Neuronal function evaluation of the MCAO mice in three groups by neurological scoring at day 3 and day 7 after the injury (*n* = 6 independent animals). *P = 0.0344, *P = 0.0344 compared between the groups of T-mPDA-Pep-Mino and T0-M0, and T0-M3 at Day 3, respectively; *P = 0.0156, *P = 0.0156 compared between the groups of T-mPDA-Pep-Mino and T0-M0, and T0-M3 at Day 7, respectively. **d** Relative expression of IL-1β, IL-6, IL-4 and IL-10 in brain tissue in different groups confirmed by ELISA assay (*n* = 5). **P = 0.0064, **P = 0.001, ****P < 0.0001,

****P < 0.0001 compared between the groups of Saline and T-mPDA-Pep-Mino for evaluation of IL-1β, IL-6, IL-4, IL-10, respectively. **e** Glial scar evaluation by immunofluorescence GFAP staining of the ischemic penumbra for in different groups. Created with BioRender.com. **f** Neurogenesis evaluation by BrdU and DCX double staining and **g** corresponding analysis of the mice brains in different groups (*n* = 5). ***P = 0.0007, ***P = 0.0004, compared between the groups of T-mPDA-Pep-Mino and Saline, and Mino. **h** Blood flow images and **i** corresponding analysis of the mice brains in different groups (*n* = 5). ****P < 0.0001, ****P < 0.0001, ***P = 0.002, ***P = 0.003, compared between the groups of T-mPDA-Pep-Mino and Saline, T-mPDA, Mino, and T-mPDA+Mino. A two-tailed P value of <0.05 was considered statistically significant. The data are presented as means ± SEM. Source data are provided as a Source Data file.

and Mino at day 3), and T-mPDA-Pep-Mino. The survival rates in three groups in Fig. 4b clearly verified the advantage of on-demand regulation ($P^* <0.05$ via survival curve comparison with Log-rank test). Neurological deficit scores of group T-mPDA-Pep-Mino at day 3 and day 7 were apparently lower than that of group T0-M0 and T0-M3 (Fig. 4c) ($P^* <0.05$ via $t$ test). These behavioral profiles proved that pathogenesis-adaptive therapy played a pivotal role in promoting the rehabilitation of brain injury. To confirm whether the recovery was resulted from on-demand microglia polarization, immunofluorescence staining of the brain slices from each group was carried on. Less signals of CD86 marker and more signals of CD206 marker were found in group T-mPDA-Pep-Mino compared with the MCAO mice with saline injection (Supplementary Fig. 26), and more pro-inflammatory signals were observed in group T0-M0 and T0-M3 (Supplementary Fig. 27). Importantly, the intracerebroventricular injection of MMP-2 inhibitor decelerated the polarization in vivo (Supplementary Fig. 28), suggesting the key role of MMP-2 to trigger the transition in accordance with in vitro study. The phenotype transition of MCAO mice treatment with the inhibitor only was also confirmed and negligible changes were found in comparison with the saline group, which excludes the positive effect from MMP-2 inhibitor itself. Thanks to the responsive drug release, the expression levels of pro-inflammatory cytokines including IL-1β and IL-6 in brains were decreased from 2.5-fold, 3.2-fold to 0.8-fold, 1.1-fold, respectively (Fig. 4d). In the meantime, higher levels of anti-inflammatory cytokines including IL-4 and IL-10 were recorded in T-mPDA-Pep-Mino group.

The remarkable melioration of the inflammatory conditions could be attributed to the sequential regulation by mesoporous polydopamine and released Mino. During ischemia-reperfusion injury, an explosive increase of activated glial cells were accumulated in the brain and cause glial scar formation, disturbing the self-repair ability of brain tissues. We evaluated the expression of glial fibrillary acidic protein (GFAP), a biomarker of astrocytes, in ischemic penumbra of MCAO mice after different treatments (Fig. 4e). Compared with the sham-operated group, the GFAP expression in penumbra sections of MCAO mice was significantly increased, indicating an enhancement of the number of GFAP-positive cells; however, this abnormality was attenuated by the treatment of T-mPDA-Pep-Mino. Further evaluation of brain remodeling was conducted by double staining of 5-bromodeoxyuridinc (BrdU) and doublecortin (DCX) to validate the neurogenesis. As shown in Fig. 4f, increased fluorescent signals in the group of T-mPDA-Pep-Mino confirmed the facilitated proliferation of neuroblasts. The number of newly divided cells in the group of T-mPDA-Pep-Mino is 5.8-fold than that of the Saline group (Fig. 4g) ($P^{**} < 0.01$ compared between the groups as indicated via one-way ANOVA with Tukey's multiple comparisons test). In addition, the blood flow restoration in the ischemic brain of different groups was examined. As shown in Fig. 4h, regional cerebral blood flow of each group was also visualized by a Perfusion and Oxygenation Imager to evaluate the extent of recovery. At seven days post-injection, more than 70% of MCAO mice with saline injection died as shown in the survival profile. Even for those survived, the blood flow at infarct side is $66.8 \pm 3.2\%$ of the flow at left side, proving the failure of self-recovery. Fortunately, the administration of T-mPDA-Pep-Mino promoted the blood flow to normal ($101.8 \pm 0.9\%$), which is significantly higher than that in T-mPDA, Mino or T-mPDA+Mino. (Fig. 4i) ($P^{***} < 0.001$, $P^{****} < 0.0001$ compared between the groups as indicated via one-way ANOVA with Tukey's multiple comparisons test). These results convincingly demonstrated our nanosystem could provide an on-demand regulation strategy for microglia polarization from ischemia-associated phenotype to recovery-associated phenotype, which together with ROS scavenging at the initial stage contributes to sequential therapy of ischemic stroke.

## In vivo evaluation of biosafety and biocompatibility of the treatment

Biosafety and biocompatibility evaluation of the treatment was performed on healthy mice. First, the rotarod test assessing the motor coordination revealed no negligible difference between the control group and the group treated with T-mPDA, Mino, or T-mPDA-Pep-Mino in eight trials (Fig. 5a). Open-field test was conducted to validate whether locomotor activity and anxiety-like behavior of the rodents were affected by the treatment. At 24 h postinjection, the total running distance and the mean speed of four groups have no significance and they shared similar running routes and heatmap as shown in Fig. 5b−d. Fluorescence staining of brain cells including neurons, astrocytes, and microglia in different regions including CA1, CA3, and cortex was performed to validate its cytotoxicity in the brain (Fig. 5e and Supplementary Figs. 29−31). No obvious cell loss was observed in each area. Immunohistological staining of major organs including heart, liver, spleen, lung, and kidney was conducted (Supplementary Fig. 32). Compared with the control group, the treatment of T-mPDA-Pep-Mino caused barely any difference, indicating its imperceptible damage on organs. Furthermore, a blood chemistry study was conducted to reveal the blood compatibility (Fig. 5f). No difference was found in the analysis of total protein, albumin, globulin, total bilirubin, and blood urea nitrogen (BUN) at 24 h post-injection. But it should be noted that alanine aminotransferase (ALT) and aspartate aminotransferase (AST) in the group of Mino (10 mg/kg) was increased compared with the control group ($P = 0.4249$ and $P = 0.2768$ compared with the saline group via paired $t$ test in ALT and AST study, respectively), indicating the potential liver toxicity of Mino[29]. However, at the same dosage, the group of T-mPDA-Pep-Mino has no discernable impact on hepatic functions, further emphasizing the importance of targeted delivery and on-demand drug release.

## Discussion

Ischemia reperfusion initiates rapid and dynamic pathological alterations in the injured brain. At the initial stage after reperfusion, excessive ROS accumulation leads to severe brain damage, and afterwards activated microglia act as a double-edged sword during the transition between pro-inflammatory and anti-inflammatory phenotype. Traditional strategies including administration of anti-inflammatory agents and/or neurotrophic factors could not respond to the dynamic changes of neuroinflammation, resulting in untimely treatment. In this work, a pathogenesis-adaptive strategy of sequential therapy based on T-mPDA-Pep-Mino is anticipated to take full advantages of mesoporous polydopamine and the MMP-2 responsive motif, contributing to synergistic therapy at both acute and chronic stages in ischemia-reperfusion injury. The intrinsic property of radical scavenging of polydopamine strengthened by mesoporous structure could attenuate the ROS-induced oxidative damage at the first stage. Along with the development of the inflammation cascade, microglial MMP-2 secretion as a marker of overactivation is used to trigger the drug release for on-demand regulation. This pathogenesis-adaptive therapeutic strategy significantly improved the survival rate and facilitated the brain recovery, overcoming the limits of untimely regulating of pathological evolution. Therefore, our work offers a proof-of-concept study to address the issue of post-stroke pathological evolution, serving as a complementary strategy for ischemia stroke therapy.

To construct the nanosystem, mesoporous polydopamine was prepared as functional cores by taking advantages of its intrinsic nature of ROS scavenging and meso-structure. Further incorporation of responsive drug release motif and brain-targeted peptide endowed the nanosystem with possibility of enhanced stroke therapy. Fortunately, the surface modifications have limited effects on the ROS scavenging of polydopamine as phenolic groups are retained after the Schiff's base

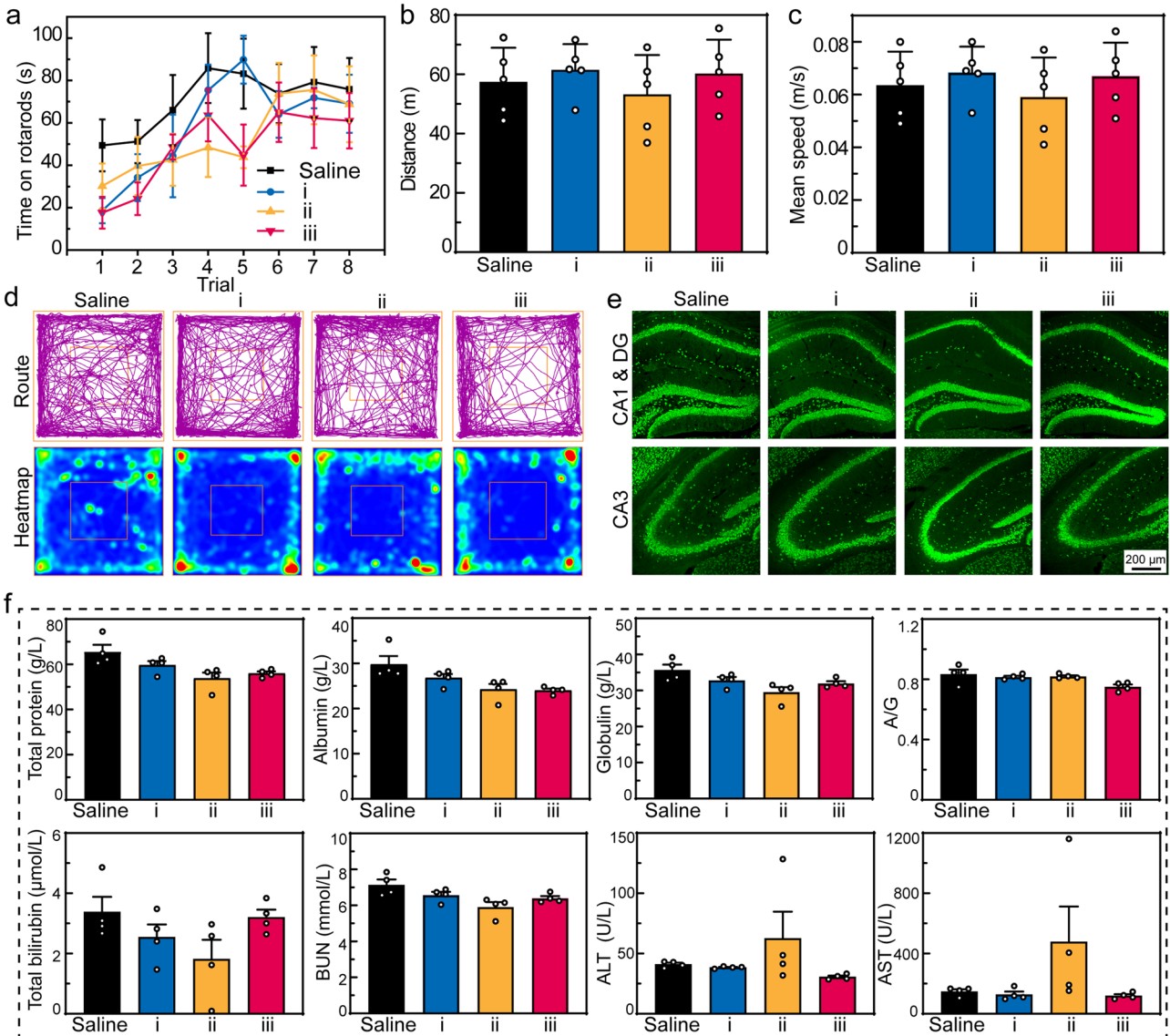

**Fig. 5 | In vivo biosafety and biocompatibility evaluation of the nanosystem.** Saline, i, ii, and iii represent the groups of control, T-mPDA, Mino, and T-mPDA-Pep-Mino, respectively ($n = 5$ independent animals). **a** Rotarod analysis of the mice at 24 h post-treatment. **b** Distance and **c** mean speed of the mice were recorded over a set time of 15 min in four groups. **d** Typical motion route (up) and heatmap (down) in an open-field test in four groups. **e** Representative immunofluorescent images of neurons in CA1, DG, and CA3 of brain slices in four groups. Neurons were stained with NeuN (green). **f** Blood chemistry analysis of the mice including total protein, albumin, globulin, total bilirubin, BUN, ALT, and AST ($n = 4$ independent animals). Mice were treated once a day for one week. The data are presented as means ± SEM. Source data are provided as a Source Data file.

reaction between thiols and polydopamine[30]. In on-demand regulation, sensitive drug release of Mino played a crucial role in microglial polarization. The drug-releasing profiles suggested a concentration-dependent mechanism of the nanosystem in response to MMP-2 and the releasing rate was saturated when the concentration is higher than 0.1 µg/mL. Previous studies on both rodent models and human brain revealed elevated MMP-2 activity at a concentration range from 0.1 to 0.2 µg/mL after infarction[22,23]. Such response threshold ensured the sensitive and accurate drug release upon microglial overactivation. Importantly, the released Mino with remaining ligand which covalently bound with 9-amino group of Mino maintained the therapeutic effects[31]. Meanwhile, in vitro studies on radical scavenging of the nanosystem provided solid evidence that T-mPDA-Pep-Mino has neuroprotective effects, which not only resulted from ROS scavenging, but also were improved by the regulation of microglial polarization and the microglia-neuron crosstalk. Characterizations of morphological, transcriptional, metabolic, autophagic, and functional states of

microglia indicated that they could respond to the T-mPDA-Pep-Mino treatment with a transition from ischemia-associated phenotype to recovery-associated phenotype. To validate the therapeutic efficacy in vivo, we investigated the performance of T-mPDA-Pep-Mino on a MCAO model. After injection, the nanosystem was able accumulate at the ischemic brain due to the brain-targeted functionalization. Behavior study and infarct volume analysis confirmed the in vivo protective effects. As known, Mino is one of the main anti-stroke medications and the ROS scavenging of polydopamine has also been reported recently. However, the treatment of T-mPDA-Pep-Mino is not a simple combination of polydopamine and Mino. As the pathological alterations in different patients varied from one to another, it is difficult to define the optimal administration time. Also, inconsistent pathological progression in different brain sections (e.g., ischemic core, penumbra, and unaffected region) made precision treatment a Herculean task in stroke therapy. We compared behavior results of T-mPDA-Pep-Mino with the treatments of T0-M0 and T0-M3. It was found that both

administration of Mino at day 0 and day 3 showed less survival rate and lower scores than that of T-mPDA-Pep-Mino. Further comparative study including cytokines production analysis, GFAP staining, neurogenesis and brain recovery evaluation indicated that sequential strategy has higher therapeutic efficacy than mono-therapy and combined therapy. These findings convincingly proved that on-demand drug release rather than drug administration at a fixed time is preferred. Thanks to the sequential treatment at both acute and later stages, the inflammatory condition in the ischemic brain was continuously improved.

In summary, we reported a pathogenesis-adaptive polydopamine nanosystem for sequential regulation of neuroinflammation after ischemia reperfusion. Taking advantages of the mesoporous structure, ROS scavenging nature, MMP-2 responsive drug release and brain-targeted drug delivery, the nanosystem was able to fight against the pathological alterations at the dynamic stages of reperfusion-induced injury. Under such a sequential treatment, the stroke animal models were demonstrated to have a higher survival rate and improved brain recovery. Together with the satisfactory biosafety profiles, our nanosystem as a pathogenesis-adaptive strategy for sequential and on-demand anti-stroke treatment provides a promising approach to ischemia-reperfusion injury, and sheds light on further investigation of other CNS disorders.

## Methods

All animal experiments were performed in compliance with the Laboratory Animal Welfare and Ethics Committee of Zhejiang Chinese Medical University and the National Institutes of Health Guide for the Care and Use of Laboratory Animals.

### Chemicals and materials

Dopamine hydrochloride, Pluronic F127 block copolymer (F127), 2-(N-Morpholino) ethanesulfonic acid, 4-Aminophenylmercuric acetate were purchased from Sigma-Aldrich. 1, 3, 5-trimethylbenzene was purchased from Tokyo Chemical Industry Co., Ltd. MMP-2 responsive peptide (Ac-CSSSGPLGIAGQSSS, 1.4 kDa) and RAP-12 brain-targeting peptide (EAKIEKHNHYQK) were synthesized by Top-peptide Co., Ltd and Chinese peptide Company, respectively. The sulfo-cyanine 5.5 NHS ester and thiol-PEG NHS ester (Mw: 5 kDa) was obtained from Yare Biotech, Inc. The Dulbecco's modified eagle's medium (DMEM), fetal bovine serum, and 0.25% (w/v) trypsin solution were purchased from Gibco BRL (Gaithers-burg, MD, USA). Primary antibodies for immunostaining including CD86 (1:400; ab119857; abcam), CD206 (1:400; 24595; CST), 5-bromo-2′-deoxyuridine (Brdu, 1:400; ab1893; abcam), Doublecortin (DCX, 1:800; 4604; CST), Neurons (NeuN, 1:400; MABN140; Millipore), GFAP(1:200; 3670; CST), Iba1(1:200; 17198; CST) were purchased as indicated. Primary antibodies for Western blot including Bax(1:1000; A19684; Abclonal), Bcl-2 (1:1000; A19693; Abclonal), β-actin (1:5000; AC038; Abclonal) were purchased as indicated. Secondary antibodies for immunostaining, including Goat Anti-Rat Alexa Fluor 488 (1:800; ab150157; abcam), Donkey Anti-Rabbit Alexa Fluor 594 (1:800; ab150108; abcam), Donkey Anti-sheep Alexa Fluor 647 (1:400; ab150179; abcam), Donkey Anti-Rabbit Alexa Fluor 488 (1:400; 711-545-152; Jackson) were purchased as indicated. Secondary antibody for Western blot including Goat Anti-Rabbit IgG (H + L) HRP labeled secondary antibody (1:5000; AS014; Abclonal) was purchased as indicated. The MMP-2 inhibitor II was purchased from GLPBIO. The cell counting kit-8 was purchased from Biosharp. Fluo-8 AM calcium staining kit was purchased from GOYOO Biotech Co., Ltd. Calcein-AM/PI Double Stain Kit was purchased from Yeasen Biotechnology. Ultrapure water (18.2 MΩ·cm) was purified using a Sartorious AG arium system and used in all experiments. All other chemicals and bioreagents were purchased from Sigma Aldrich unless otherwise declared.

### Characterization

Scanning electron microscopy observations were conducted on an FESEM (SU-8010, Japan). Transmission electron microscopy images were obtained on a JEOL electron microscope (JEM-1400, acceleration voltage 120 kV). UV-Vis-NIR absorbance of the materials were recorded on a Cary5000 spectrometer and fluorescence spectroscopy was performed on an F-4700 spectrofluorometer (Horiba). Thermogravimetric analysis of the lyophilized materials was performed on a Q50 TGA analyzer. The temperature elevation was conducted from room temperature to 700 °C at a scanning speed of 10 °C/min with a $N_2$ flow of 40 mL/min. Hydrodynamic diameter and zeta potential analysis of the materials was performed by a Malvern nanosizer (NANO-ZS). The nanomaterials at a concentration of 0.2 mg/mL in a neutral buffer was prepared for dynamic laser scanning analysis. Nuclear Magnetic Resonance (NMR) spectroscopy was conducted on an Avanc III 600 MHz Digital NMR Spectrometer (Solvent: DMSO-d6). Molecular weight determination of the materials was performed by matrix-assisted laser desorption ionization-time of flight (MALDI-TOF, ultra-flextreme, Bruker). Cell and tissue observation was conducted on Carl Zeiss LSM880 laser scanning confocal microscopy and Leica DMIL LED inverted fluorescence microscope. Figures 2i, 3a, 4a, and 4e are created with BioRender.com.

### Preparation of nanoparticles

Mesoporous polydopamine was synthesized via a mico-emulsion method[32,33]. In brief, a mixture of F127 (50 mg) in ethanol (1.5 mL) and dopamine hydrochloride (15 mg) in water (1.5 mL) was prepared and followed by addition of 1, 3, 5-trimethylbenzene (30 μL). After three minutes ultrasonication, 120 μL of ammonium hydroxide was added to trigger the reaction. The raw products could be obtained by centrifugation (×10,000 g, 10 min). 1, 3, 5-trimethylbenzene and F127 polymer can be removed by ethanol and water. The purified nanoparticles could be stored at 4 °C for further use.

Non-mesoporous polydopamine was synthesized by self-polymerization of dopamine in bicine buffer (pH = 8.5). The nanoparticle size is determined by the amount of dopamine or reaction time. For instance, in 10 mL of 0.5 M Bicine buffer, 4 mg of dopamine hydrochloride was added and was allowed to polymerize under continuous stirring. The products could be obtained by centrifugation (× 10000 g, 10 min) and washing.

Drug loading was performed by using MMP-2 responsive peptide as a linker. Prior to the loading, 20 mg of Pep was dissolved in 2 mL of 0.1 M MES buffer (pH = 5.2). After adding 21.2 mg of EDC and 78 mg of sulfo-NHS, carboxyl groups of the Pep were activated and then reacted with 29.2 mg of 9-amino minocycline for 24 h. The Pep-Mino conjugate was purified by dialysis (MWCO: ~1000 Da) and lyophilized for 48 h. Then, the Pep-Mino conjugation was added excessively in the alkaline solution of mesoporous polydopamine (2 mg/mL) and was allowed to react 24 h under magnetic stirring. The drug-loaded nanoparticles were obtained by centrifugation (×10000 g, 20 min) and stored at 4 °C for further use. Drug loading capacity $β$ was calculated by the equation $β = (C_0 - C_s)/C_n$, where $C_0$ is the amount of the drug added at the initial stage, $C_s$ is the amount of the drug in the supernatant that measured by high-performance0 liquid chromatography with C18 column by using a UV detector (The mobile phases comprise ammonium acetate buffer, acetonitrile, and methanol), and $C_n$ is the amount of the mesoporous polydopamine added at the initial stage.

Brain targeting functionalization of the nanoparticles was conducted by surface modification of PEGyated RAP-12. In brief, 10 mg of PEG-NHS ester polymer (MW: ~5000 Da) was used for conjugation with 66 mg of RAP-12 in 2 mL of 0.2 M $Na_2CO_3$ solution. After 24 h of reaction, the raw products were purified by dialysis (MWCO: ~3500 Da) against PBS buffer. Then, the purified PEGyated RAP-12 was conjugated with drug-loaded nanoparticles to obtain the targeted nanosystem in bicine buffer (pH = 8.5).

## Cell culture and cytotoxicity study

To investigate the nanosystem in vitro, SH-SY5Y, and BV-2 cell lines were cultured in DMEM supplemented with 10% FBS, 100 IU/mL penicillin and 100 mg/mL streptomycin sulfate. The cell lines were cultured in a 5% $CO_2$ incubator at 37 °C. All cellular studies were performed by utilizing the cells at their logarithmic growth phase. The cellular biocompatibility was evaluated by CCK-8 assay kit. The SH-SY5Y cells at a cell density of $1 \times 10^4$ cells per well were cultured with T-mPDA-Pep-Mino nanoparticles at different concentration up to 800 μg/mL and incubated for 24 h. Then, 10 μL of CCK-8 solution was added into each well and incubated for another 4 h. Absorbance of each test solution which was recorded by a microplate reader at 450 nm and 570 nm was used to evaluate the cell viability.

## In vitro evaluation of neuroprotective effects

To evaluate the ROS scavenging capability of the materials, an ROSup assay kit (Beyotime, S0033M) was used and SH-SY5Y cells were cultured for the evaluation. In brief, the SH-SY5Y cells were treated with ROSup probe (compound mixture) according to the manufactory instruction, and then either mesoporous polydopamine or non-mesoporous polydopamine nanoparticle was added. After 12 h incubation, the ROS probe (DCFH-DA) was added in each group at an advised concentration. Finally, the cells could be further observed by fluorescence microscope. Further cellular evaluation of the neuroprotection was conducted on an oxygen-glucose deprivation (OGD) model by using SH-SY5Y cells. The cells were cultured in a glucose-free DMEM and placed in a hypoxia incubator (95% $N_2$, 5% $CO_2$). After OGD for 4 h, the cells were transferred to a 5% $CO_2$ incubator with normal oxygen supply and cultured with different treatments (saline, T-mPDA, Mino, T-mPDA-Pep-Mino) at a Mino concentration of 10 mg/mL in DMEM (without FBS) for 24 h. Afterwards, the treated cells were available for further evaluation including TUNEL staining (TUNEL Apoptosis Detection Kit, Alexa Fluor 488, Beyotime), western blot (Bax: Bax Rabbit mAb, Abclone; Bcl-2: Bcl-2 Rabbit mAb, Abclone), calcium staining (Fluo-8 AM calcium staining kit, AAT Bioquest), and flow cytometry (Annexin V-FITC Apoptosis Detection Kit, BD 556547). For Annexin V-FITC/PI staining: The cells were harvested and stained with Annexin V-FITC and PI for 15 min. After washed three times with PBS, the cells were resuspended in PBS followed by recording immediately using a flow cytometer (Beckman, coulter). And then analyzed using CytExpert software based on 10,000 gated events. For calcium staining by Fluo-8 AM, the cells were collected and stained using Fluo-8 AM staining kit, and then subjected to flow cytometry for intracellular calcium detection. The fluorescence intensity of over 10,000 cells was measured with flow cytometer (Beckman, coulter), and analyzed with CytExpert software. The gating strategy was shown in Supplementary Figure 33.

## In vitro evaluation of regulation of microglial polarization

To evaluate the responsive drug release and its effects on microglial polarization, BV-2 cells were cultured and treated by OGD for 4 h and reperfusion for 24 h for further cellular study. In each group, the cells were treated by either saline, T-mPDA, Mino, or T-mPDA-Pep-Mino at a Mino concentration of 10 μg/mL. At 24 h post-treatment. The microglial phenotype transition in different groups were investigated by immunofluorescence staining of CD86 and CD206. The cells for immunostaining were fixed by 4% paraformaldehyde for 15 min. After PBS washing, the cells were treated by 0.1% Triton X-100 in PBS for 20 min. Then, the slides were incubated with a solution of donkey serum (5%) for 2 h and added with CD86 or CD206 antibody. After incubated at 4°C overnight, the cells were slightly washed by PBS for three times and were used for further immunofluorescence staining. Morphological changes of BV-2 cells were observed in a bright field by a Leica inverted fluorescence microscope. A transwell study with co-cultured cells was used to evaluate the neuroprotective effects of microglial polarization. In brief, BV-2 cells were cultured at the upper chamber and SH-SY5Y cells were seeded at the down chamber. After OGD for 4 h, different treatments were given at the upper chamber and the cells were re-supplied with oxygen. At 24 h post-treatment, the cell medium at the down chamber was collected for ELISA tests performed by Mouse ELISA Kit of TNF-α, IL-6, IL-10, and TGF-β1. For cellular viability evaluation of SH-SY5Y cells at the down chamber, Live/Dead assay kit was used to study the neuroprotective effects in each group after the treatments. For transcriptome analysis, BV-2 cell line was cultured and treated under different conditions (Normoxia, OGD, T-mPDA, Mino, T-mPDA-Pep-Mino) Each test was conducted for three repetitive times and >$1 \times 10^6$ cells were collected for one sample. The collected cells were treated by 1 mL Trizol reagent. Further transcriptome sequencing was performed by Beijing Tsingke Biotech Co., Ltd.

## In vivo mice model for therapy evaluation

A middle cerebral artery occlusion (MCAO) model was established by using male C57BL/6 mice at age of 6-8 weeks for therapy evaluation. Mice were randomly divided into five groups (sham, saline, T-mPDA, Mino, T-mPDA-Pep-Mino). For later four groups, mice were treated by MCAO process. The experiments were conducted under isoflurane anesthesia. Prior to the surgery, a fiber-optic probe was anchored onto the brain skull (2-mm caudal to bregma and 6-mm lateral to midline) to the right middle cerebral artery (MCA). Then, a VMS-LDF2 laser doppler flowmetry (Moor Instruments Ltd, UK) was used to record cerebral blood flow (CBF) at the MCA region. A 6-0 nylon monofilament suture with blunted tip and 1% poly-L-lysine pre-treatment was introduced in internal carotid (10 mm) to occlude blood flow at MCA. Animals were excluded from the experiments if the recorded CBF is higher than 20% of the normal level. To prevent hypothermia, a temperature maintenance instrument (RWD Life Science) was used during surgery, and reperfusion was allowed after 60 min of ischemia by gently withdrawing the monofilament. Then the mice in each group (saline, T-mPDA, Mino, T-mPDA-Pep-Mino) were intraperitoneally injected with different drugs at a Mino concentration of 10 mg/kg at the onset of reperfusion. The mice in sham group were conducted with the same procedures but without monofilament blocking and drug administration. To evaluate the difference between of combined therapy and sequential regulation, the MCAO mice were treated by T0-M0, T0-M3, or T-mPDA-Pep-Mino. Mice in T0-M0 group are injected with T-mPDA at day 0 and Mino at day 0, and mice in T0-M3 group are injected with T-mPDA at day 0 and Mino at day 3. To evaluate the efficacy of MMP-2-responsive drug release, the mice were intracerebroventricularly injected with 2 μL of 1.8 μM ARP-100. To evaluate the advantage of sequential therapy, combined therapy (T-mPDA+Mino, administration of a mixture of T-mPDA and Mino at the same carrier and drug concentrations with T-mPDA-Pep-Mino) was also conducted in some studies as indicated in the manuscript. After the therapy, animals were separately housed. All mice were maintained under a 12 h light-dark cycle (light on from 8:00 a.m. to 8:00 p.m.) with ad libitum access to food and water. The ambient temperature is 20-26 °C and the humidity is 40-70%. All behavior tests were conducted between 13:00-17:00.

## In vivo evaluation of therapeutic performance

To investigate the intracerebral distribution, the brain sections were stained by 2,3,5-triphenyltetrazolium hydrochloride (TTC, 0.25%) at 24 h post-reperfusion. Infarct volumes of the animal models were measured by ImageJ and determined by an indirect method that corrects for edema. The proportion of infarct volume = (contralateral uninfarct area - ipsilateral uninfarct area)/total contralateral uninfarct area × 100%. Neurological deficit scoring method was used for behavior evaluation at 24 h post-surgery as follows: 0, no deficit; 1, flexion of contralateral forelimb upon lifting of the whole animal by the tail; 2,

circling to the contralateral side; 3, falling to contralateral side; 4, no spontaneous motor activity; 5, death. At day 3 after the surgery, mice were intravenously injected with 0.5 % EB solution at a volume of 10 μL/g body weight 2 h before sacrifice. Mice then were transcardially perfused with PBS and 4% (w/v) paraformaldehyde (PFA). Ipsilateral hemispheres were collected and homogenized in PBS, followed by centrifugation (×12,000 g, 15 min). The supernatant was collected and added with acetone at a ratio of 3:7. After 24 h incubation at room temperature, the EB absorbance was measured using a spectrophotometer at 620 nm.

### Immunofluorescence staining
The treated mice were transcardially perfused with PBS and 4% (w/v) PFA. Brain sections were collected and fixed by 4% (w/v) PFA at 4 °C overnight. After 48 h of sucrose (30%) incubation, coronal sections of brain tissue 25-μm thickness could be obtained by using an NX50 cryostat (Thermo Fisher). Then, the brain slices were permeabilized with Triton X-100 (0.1% in PBS) for 15 min. After treated by donkey serum (5% in PBS) for 2 h at room temperature, brain slices were incubated with anti-CD86/CD206 antibody (1:400), anti-BrdU antibody (1:400), anti-DCX antibody (1:800), anti-NeuN antibody (1:400), anti-GFAP antibody (1:200), anti-Iba-1 antibody (1:200) at 4 °C overnight and washed with PBS for three times, and then incubated with Alexa Fluor-conjugated secondary antibody (1:400) for 2 h at room temperature. DAPI was used as a nuclear stain. All sections at a similar coronal position are observed under a Leica Virtual Digital Slice Scanning System.

### Statistical analysis
The data are presented as means ± SEM, and a two-tailed $P$ value of <0.05 was considered statistically significant in all cases. A one-way ANOVA with Tukey's multiple comparisons test was used to compare the results of ELISA, infarct area, behavior study and other tests as indicated. Paired $t$-test was used to compare the results of fluorescence study, blood chemistry study, EB staining and other tests as indicated. Survival curve comparison was used to compare the results of survival rates with Log-rank test.

### Reporting summary
Further information on research design is available in the Nature Portfolio Reporting Summary linked to this article.

## Data availability
All data needed to evaluate the conclusions in the paper are present in the paper and/or the Supplementary Materials. The full image dataset is available from the corresponding author upon request. Source data are provided with this paper.

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

## Acknowledgements
This project was supported by grants from the National Natural Science Foundation of China (82273861 to D.W., 82003666 to D.W., 82273903 to Yanrong Z.), Young Elite Scientists Sponsorship Program by China Association for Science and Technology (YESS20220139 to D.W.), National Key R&D program of China (2020YFA0803902 to C.Z.), and Natural Science Foundation of Zhejiang Province (LD22H310003 to C.Z.). We appreciate the support from Public Platform of Medical Research Center, Academy of Chinese Medical Science, Zhejiang Chinese Medical University.

## Author contributions
D.W. and Z.C. conceived and designed the project. D.W., J.Z., Yanrong Z., and Yuyi Z. performed the experimental work. D.W., J.Z., Yanrong Z., Yuyi Z., Q.Z., Z.Z., X.C., Q.C., Y.R., and Y.W. analyzed the data. D.W. and Z.C. wrote the manuscript. All authors discussed the results and contributed to the final manuscript.

## Competing interests
The authors declare no competing interests.
