## [Peer Review File · Nature Communications]

Reviewers' Comments:

Reviewer #1:

Remarks to the Author:

The author prepared mesoporous PDA as inner core to conjugate sensitive peptide and therapeutic agent for achieving the effects of responsive drug release and brain targeting. The paper was well organized and the authors have carried out numerous experiments at the cellular and animal levels to support the hypothesis. A novel treatment concept was also proposed for neuroprotection after ischemia stroke.

1 In Fig 1-A, the Pep-Mino was grafted on the inner surface of mPDA, not the outer surface.

Actually, such grafting reaction is not specific and the reaction proceeds more easily on the outer surface than the inner surface. Have the authors confirmed the correctness of this schematic illustration?

2 Normally, breakage of responsive or sensitive linker should release free drugs to obtain expected efficacy. In this work, when degradable peptide was disrupted by MMP2, Mino with amino acid residues may be released. Please provide related references or test results to prove the consistency of therapeutic effects of Mino and Mino with amino acid residues.

3 ".....drug loading capacity of the nanosystem was determined to be 104.3 mg/g, which is consistent with the thermogravimetric analysis results (Figure 1E and S4)." Please explain how to get this?

4 Many reports indicated that PDA nanoparticles were pH-, and ROS sensitive, and rapid drug release could be triggered with lower pH value and H₂O₂. But this work showed no significant difference of dissolution curves between normal control group and test groups in Fig s7. Please explain these results.

5 In addition, symbols in legend were different with that on the curves in Fig s7-B (red circle indicated). The author should check.

6 PDA is the key factor to achieve ROS scavenging. In Fig 1-I, mPDA showed better ROS scavenging effect compared to PDA owing to larger surface area. On the other hand, when Mino and brain targeting peptide ligand were conjugated on the mPDA, the active reaction area reduced. However, in Fig 1-I, T-mPDA-Pep-Mino group showed excellent ROS removal capability compared to other test groups. Please explain this.

7 In Fig s8, symbols of significant difference were presented but groups for comparison were not indicated.

8 *In vivo* disposition behaviors of systemic circulation and tissue retention of nanosystem are important, especially for designing of active targeted drug carrier. It was not sufficient to qualitatively study the active brain targeting effects using fluorescent probe labelled-nanoparticles. To prove the merits of this designed nanosystem, quantitative analysis of *in vivo* pharmacokinetics and biodistribution for comparison were needed.

9 To exert the synergistic treatment effects of constructed nanosystem T-mPDA-Pep-Mino, results comparisons of significant difference among test groups should be indicated in pharmacodynamic studies (eg. Fig 2-J, Fig 3-D,E, Fig 4-I, Fig s18 etc.).

10 Too many abbreviations were used in this work (eg. PDA, mPDA, T-mPDA, Pep, Mino, T-mPDA-Pep-Mino..... etc.). For convenience, it is recommended to summarize and list all abbreviations in the supporting materials.

11 In supporting materials, the author should double check the writing accuracy. For instance, "In brief, Briefly, 50 mg of F127 were dissolved in 1.5 mL of ethanol, 30 μ L of 1, 3, 5-trimethylbenzene, 15 mg of dopamine hydrochloride."

12 Centrifugation parameters (rpm, time) should be listed.

13 Unit should be written in uniform format. (mL-1, /mL etc.)

Note from the Editor: please find attached additional comments from Reviewer #1.

1 symbols in legend were different with that on the curves in Fig s7-B (red circle indicated).

Reviewer #2:

Remarks to the Author:

This is a well-designed and elegant proof of concept in vivo and in vitro study finally demonstrating on demand drug release after experimental ischemic stroke. Using mPDA-Pep-Mino, a pathogenesis-adaptive drug delivery system, the authors conclusively show reduction of ROS toxicity in the acute reperfusion phase in a stroke model in mice followed by an on demand induction of a protective and recovery enhancing microglia phenotype. Brain protection is shown to be associated with enhanced neurological outcome. This sequential strategy is a very interesting and promising novel therapeutic approach.

This reviewer has only one criticism. Fig. 4F shows increased proliferation in the mPDA-Pep-Mino group. Whether these cells are really neuroblasts would require double staining with a respective neuronal marker.

Reviewer #3:

Remarks to the Author:

The study by Wu et al. details the use of a drug delivery system that is believed to be beneficial for treatment of ischemic stroke. The drug system is comprised of a reactive oxygen scavenger polydopamine and a MMP2 responsive minocycline. The minocycline has been constructed so that it will only be released in the presence of MMP2 which is a very interesting paradigm to target therapies to a specific insult. The authors demonstrate that the drug compound is protective when given 1 hour after MCAO. Additionally, the authors suggest the benefit of the model is the induction of M2 anti-inflammatory microglia and loss of M1 inflammatory microglia. While the drug set up is intriguing, there are several concerns with the interpretation of the results.

- 1) A major concern is the use of the M1 vs M2 nomenclature for microglia. This is a concept that has become very out of date for the field. Microglia are incredibly complex and cannot be captured by simple M1 vs M2 statements. Many microglia share overlapping inflammatory AND anti-inflammatory markers which makes this sort of dichotomous interpretation challenging. The authors need to do a more comprehensive characterization of the glial response if they want to make a statement about glial phenotypes.
- 2) The idea of the sequential/on-demand treatment is not entirely clear. It would seem that the drug will be activated as soon as it is injected after ischemic condition as there should be lots of MMP already present. It's unclear how this is different than just injecting minocycline or polydopamine directly. This should be better clarified. Perhaps some additional experiments showing MMP levels during the course of the ischemic event would be helpful to better understand the timeline for the drug activation.
- 3) The authors suggest that although minocycline and polydopamine on their own have positive effects, the combo designed drug is more beneficial. However, is the designed drug better than just simply injecting both polydopamine and minocycline at the same time? It isn't surprising two beneficial drugs would have at least an additive effect. More details are needed to demonstrate the designed drug is better than the simple combination treatment.

Point-by-point Response to Reviewers' Comments

Reviewer #1

The author prepared mesoporous PDA as inner core to conjugate sensitive peptide and therapeutic agent for achieving the effects of responsive drug release and brain targeting. The paper was well organized and the authors have carried out numerous experiments at the cellular and animal levels to support the hypothesis. A novel treatment concept was also proposed for neuroprotection after ischemia stroke.

Response: We thank the reviewer for the kind consideration of and constructive comments on our manuscript. We have carefully revised the manuscript and provided the point-by-point response below in blue. The changes in the revised manuscript have been highlighted. We hope these changes could strengthen our manuscript.

1 In Fig 1-A, the Pep-Mino was grafted on the inner surface of mPDA, not the outer surface. Actually, such grafting reaction is not specific and the reaction proceeds more easily on the outer surface than the inner surface. Have the authors confirmed the correctness of this schematic illustration?

Response: Thank you for the comments. We agree that this reaction is not specific and the reaction could proceed on both outer and inner surface. To demonstrate that the drug molecules could anchor on the inner surface of mPDA, we constructed a fluorescence quenching model. mPDA was coated on gold nanoparticles, giving core-shell nanoparticles (Au@mPDA) with the similar particle size and pore size with the mPDA nanoparticles (Figure 1A). Then, green fluorescent protein (GFP) was used as a distance sensor because the gold nanoparticle could quench the GFP if the protein and the gold core are close enough, which could trigger the quenching. For comparison, non-mesoporous polydopamine-coated gold nanoparticles (Au@PDA) were also prepared (Figure 1B). After incubation, the fluorescence of GFP with Au@mPDA solution significantly decreased (Figure 1C). In contrast, the GFP incubated with non-mesoporous Au@PDA solution showed limited fluorescence reduction because the protein absorption on outer surface of polydopamine could hardly cause any quenching, which is also accordance with our previous report (*Biomaterials*, 2020, 238, 119847). This phenomenon suggested that the drug of interest could get into the inner mesopores. Herein, the reason that the GFP was used as the sensor instead of fluorescent dyes because polydopamine was able to quench most small-molecule dyes via π - π stacking, thus failing to distinguish between outer and inner surface. The β -sheet strand of GFP could prevent the fluorophore from polydopamine quenching (not gold-induced quenching). We believe that the drug of interest in this work (HS-Pep-Mino) which have lower molecular weight and smaller three-dimensional structure than GFP are easier to anchor onto the inner surface. Further, the drug loading was finished before the surface modification of targeting ligands. Together with the evidence of high drug loading capacity, we drew the schematic illustration as shown in the manuscript.

Figure 1. TEM images of (A) mesoporous polydopamine-coated gold nanoparticles (Au@mPDA) and (B) non-mesoporous polydopamine-coated gold nanoparticles (Au@PDA). (C) Fluorescence spectrum of different solutions including GFP (green triangle), GFP incubated with Au@mPDA (brown circle), and GFP incubated with Au@PDA (orange circle) at the same GFP concentration (0.2 mg/mL).

2 Normally, breakage of responsive or sensitive linker should release free drugs to obtain expected efficacy. In this work, when degradable peptide was disrupted by MMP2, Mino with amino acid residues may be released. Please provide related references or test results to prove the consistency of therapeutic effects of Mino and Mino with amino acid residues.

Response: Thanks for raising this out. In a previous report, Kannan et al. successfully synthesized poly(amidoamine) dendrimer-conjugated minocycline and proved that the conjugation does not compromise the therapeutic effects of minocycline on microglia (Bioconjugate Chem. 2017, 28, 2874-2886). This reference has also been cited in the revised manuscript. See 2nd paragraph of Discussion section (Page 20).

3 “.....drug loading capacity of the nanosystem was determined to be 104.3 mg/g, which is consistent with the thermogravimetric analysis results (Figure 1E and S4).” Please explain how to get this?

Response: Thank you for kindly reminding us. The drug loading capacity β was calculated by the equation $\beta = (C_0 - C_s) / C_n$, where C_0 is the amount of the drug added at the initial stage, C_s is the amount of the drug in the supernatant that measured by HPLC, and C_n is the amount of the mPDA added at the initial stage. The β value was calculated to be 104.3 mg/g, which indicated that the loaded drug accounts for 9.44% mass weight of the nanosystem. Further, TGA analysis was performed. TGA results shown in Figure 2A (Figure S4 of the supplementary materials) indicated that the weight loss temperature range of Mino was 185 to 305 °C. From Figure 2B (Figure 1E of the manuscript), we could find that the weight loss of mPDA-Pep-Mino is 7.25% (185 °C: 95.13%; 305 °C: 87.88%), which is at the similar level with the previous results. However, compared with HPLC measurement, composition analysis by TGA here is a relatively rough estimation. Thus, the TGA and HPLC analysis curves are provided in the manuscript and the supplementary materials but without discussion of the calculations of TGA results. According to the comments, we have added the detailed calculation method of drug loading capacity by HPLC method in the revised supplementary materials. See 4th paragraph of Materials and methods-Preparation of nanoparticles in supplementary materials. (Page 5)

Figure 2. Derivative thermogravimetry results of mPDA, Mino, and mPDA-Pep-Mino from 100 to 700 °C.

4 Many reports indicated that PDA nanoparticles were pH-, and ROS sensitive, and rapid drug release could be triggered with lower pH value and H₂O₂. But this work showed no significant difference of dissolution curves between normal control group and test groups in Fig s7. Please explain these results.

Response: Yes, there are some PDA-based materials that have been used for fabrication of pH- and ROS-responsive systems. However, the drug encapsulation and the responsive drug release mechanisms are totally different. Our recent review (*Chem. Soc. Rev.*, 2021, 50, 4432-4483) and some important reviews from other group (*Chem. Soc. Rev.*, 2021, 50, 8319-8343) have summarized the recent stimuli-responsive PDA-enabled nanosystems and introduced the drug loading and release mechanism in detail. For pH-responsive PDA materials, it has been concluded that the drugs are loaded via non-covalent interactions including hydrophobic interactions, electrostatic interactions, and *etc.* In some cases, metal ions could be incorporated and also contribute to the pH-sensitive properties as the metal ions coordinate with the catechol groups of PDA. For ROS-responsive materials, the drugs of interest are loaded mainly through hydrogen bonds. Upon ROS-induced oxidation, the non-covalent linkage between the drugs and catechol groups were broken, which further trigger the cargo release. In conclusion, PDA-based materials are able to release the drugs in response to pH change and ROS because the drugs of interest are loaded through non-covalent interactions. In our study, minocycline was first conjugated with the peptide via EDC/NHS reaction. Then, the Pep-Mino conjugates were anchored onto the mPDA surface via the chemical reaction between thiols and mPDA. Both reactions produced covalent bindings between the drug and the materials which could hardly be interrupted by pH change and oxidative stress. Therefore, minocycline could only be released in the presence of MMP-2 which is capable of enzymatical degradation of the peptide.

5 In addition, symbols in legend were different with that on the curves in Fig s7-B (red circle indicated). The author should check.

Response: We are sorry for the mistake. The symbols in the legend of Figure S7B have been corrected in the revised supplementary materials.

6 PDA is the key factor to achieve ROS scavenging. In Fig 1-I, mPDA showed better ROS scavenging effect compared to PDA owing to larger surface area. On the other hand, when Mino and brain targeting peptide ligand were conjugated on the mPDA, the active reaction area reduced. However, in Fig 1-I, T-mPDA-Pep-Mino group showed excellent ROS removal capability compared

to other test groups. Please explain this.

Response: This is an important point by the reviewer. T-mPDA-Pep-Mino is found to have ROS scavenging property mainly because of mesoporous polydopamine with enhanced surface area. It is noteworthy that Pep-Mino and brain targeting peptide ligand were conjugated on the mPDA via the reaction between thiols and mPDA as depicted below (Figure 3). After the conjugation, the catechol groups which account for ROS removal capability of PDA are maintained (*Science*, 2007, 318, 426-430), indicating that no active site for ROS reaction is reduced. Further, the conjugated minocycline could at the meantime contribute to the ROS removal performance to some extent due to its multiple phenolic groups which also may exert direct radical-scavenging activity (*J. Neurochem.*, 2005, 94, 819-827; *J. Invest. Dermatol.*, 1986, 86, 449-453). Thus, T-mPDA-Pep-Mino should have better ROS scavenging capability, which is accordance with the experimental results. These discussions have also been added in the revised manuscript. See 2nd paragraph of Discussion section (Page 20).

Figure 3. Reaction between polydopamine and thiol-ended molecules. Reproduced from the literature (*Science*, 2007, 318, 426-430).

7 In Fig s8, symbols of significant difference were presented but groups for comparison were not indicated.

Response: The groups for comparison is indicated in the figure caption now. Thanks.

8 In vivo disposition behaviors of systemic circulation and tissue retention of nanosystem are important, especially for designing of active targeted drug carrier. It was not sufficient to qualitatively study the active brain targeting effects using fluorescent probe labelled nanoparticles. To prove the merits of this designed nanosystem, quantitative analysis of in vivo pharmacokinetics and biodistribution for comparison with were needed.

Response: Thanks for your constructive suggestions. Following the comment, quantitative in vivo pharmacokinetics study of minocycline in serum by injection of T-mPDA-Pep-Mino or direct injection at a dose of 10 mg/kg was performed. Briefly, blood samples were collected at 1, 2, 4, 8, 12, 24 h post-injection. Blood samples were incubated with 1 $\mu\text{g/mL}$ MMP-2 prior to the test and the supernatants were collected by centrifugation ($\times 10000$ g, 20 min) of the mixed samples. The serum drug concentrations of different groups were determined by HPLC method at each timepoint. One-compartment model with liner absorption was used to fit the concentration-time profiles which were shown in Figure 4A. The values of C_{max} and AUC_{0-24} of T-mPDA-Pep-Mino group are determined to be 10.0 ± 1.3 $\mu\text{g/mL}$ and 73.1 ± 3.0 $\mu\text{g/mL}\cdot\text{h}$, respectively, which are higher than that of Mino group (C_{max} : 7.7 ± 0.8 $\mu\text{g/mL}$; AUC_{0-24} : 52.0 ± 4.4 $\mu\text{g/mL}\cdot\text{h}$). Although $t_{1/2}$ of the T-

mPDA-Pep-Mino profile is 4.8 ± 0.7 h, slightly faster than that of Mino group (5.8 ± 2.0 h), in vivo imaging results of the T-mPDA-Pep-Mino continuously to accumulate at brains within the first 4 h post-injection and remained at a high level in the following three days. In vivo biodistribution of T-mPDA-Pep-Mino in major organs at 4 h revealed a high amount of nanosystems were captured and eliminated by liver and kidney, which probably due to the cellular uptake by reticuloendothelial system. These results and discussion were added in the revised manuscript. See 1st paragraph of Results-Brain protection of T-mPDA-Pep-Mino in ischemia-reperfusion injury (Page 13).

Figure 4. (A) Blood retention of T-mPDA-Pep-Mino and Mino within 24 h after the drug administration. (B) In vivo imaging of MCAO mice treated with labelled T-mPDA-Pep-Mino and (C) corresponding intensity analysis of the brain regions. (D) Ex vivo imaging of the major organs of the MCAO mice treated with labelled T-mPDA-Pep-Mino. Data are shown as the mean \pm SD ($n = 3$).

9 To exert the synergistic treatment effects of constructed nanosystem T-mPDA-Pep-Mino, results comparisons of significant difference among test groups should be indicated in pharmacodynamic studies (eg. Fig 2-J, Fig 3-D,E, Fig 4-I, Fig s18 etc.).

Response: Results comparisons of significant difference among test groups in Figure 2J, Figure 3D, 3E, Figure 4I, Figure S18, and other figures that reflect the advantages of sequential therapy are indicated in the revised manuscript. Thanks.

10 Too many abbreviations were used in this work (eg. PDA, mPDA, T-mPDA, Pep, Mino, TmPDA-Pep-Mino..... etc.). For convenience, it is recommended to summarize and list all abbreviations in the supporting materials.

Response: Thank you for this valuable comment. We have given a list of abbreviation in the revised supplementary materials for convenience. See Page 2 of supplementary materials.

11 In supporting materials, the author should double check the writing accuracy. For instance, “In brief, Briefly, 50 mg of F127 were dissolved in 1.5 mL of ethanol, 30 μ L of 1, 3, 5-trimethylbenzene, 15 mg of dopamine hydrochloride.”

Response: We are sorry for the mistake. It has been corrected. We have also revised some other grammar mistakes and typos in the revised manuscript and supplementary materials.

12 Centrifugation parameters (rpm, time) should be listed.

Response: The centrifugation parameters (rpm or \times g, time) are now given in the new version of supplementary materials.

13 Unit should be written in uniform format. (mL-1, /mL etc.)

Response: We have revised the unit that was not in uniform format in the new version of both manuscript and supplementary materials.

Reviewer #2

This is a well-designed and elegant proof of concept in vivo and in vitro study finally demonstrating on demand drug release after experimental ischemic stroke. Using mPDAPep-Mino, a pathogenesis-adaptive drug delivery system, the authors conclusively show reduction of ROS toxicity in the acute reperfusion phase in a stroke model in mice followed by an on demand induction of a protective and recovery enhancing microglia phenotype. Brain protection is shown to be associated with enhanced neurological outcome. This sequential strategy is a very interesting and promising novel therapeutic approach. This reviewer has only one criticism. Fig. 4F shows increased proliferation in the mPDAPep-Mino group. Whether these cells are really neuroblasts would require double staining with a respective neuronal marker.

Response: We appreciate your summary of the manuscript and encouraging comment. To confirm the neuroblasts, the brain slices were double stained for BrdU and doublecortin (DCX) as shown below (Figure 5). Facilitated proliferation of neuroblasts was observed after the treatments. The number of newly divided cells in the group of T-mPDA-Pep-Mino is 5.8-fold than that of the group of Saline and is higher than that in other groups, which indicated its contribution to neurogenesis and brain recovery. The merged figures and corresponding analysis were updated in Figure 4F and 4G of the revised manuscript. See Figure 4 of the manuscript (Page 17).

Figure 5. Immunofluorescence staining of brain slices from the MCAO mice in different groups.

Reviewer #3

The study by Wu et al. details the use of a drug delivery system that is believed to be beneficial for treatment of ischemic stroke. The drug system is comprised of a reactive oxygen scavenger polydopamine and a MMP2 responsive minocycline. The minocycline has been constructed so that it will only be released in the presence of MMP2 which is a very interesting paradigm to target therapies to a specific insult. The authors demonstrate that the drug compound is protective when given 1 hour after MCAO. Additionally, the authors suggest the benefit of the model is the induction of M2 anti-inflammatory microglia and loss of M1 inflammatory microglia. While the drug set up is intriguing, there are several concerns with the interpretation of the results.

Response: Thank you for the comments and suggestions on the manuscript. We have carefully revised the manuscript and provided the point-by-point response below.

1) A major concern is the use of the M1 vs M2 nomenclature for microglia. This is a concept that has become very out of date for the field. Microglia are incredibly complex and cannot be captured by simple M1 vs M2 statements. Many microglia share overlapping inflammatory AND anti-inflammatory markers which makes this sort of dichotomous interpretation challenging. The authors need to do a more comprehensive characterization of the glial response if they want to make a statement about glial phenotypes.

Response: Thank you for this valuable comment. Here, we constructed a drug delivery system that confers anti-inflammatory effect with response to intrinsic inflammatory stress. Its anti-inflammatory effect was confirmed by reduced pro-inflammatory cytokine production and decreased CD86+/CD206- cells in the brain. Nevertheless, we greatly agree with the reviewer's point that the phenotypes of microglia are incredibly complex. Thus, we soften our statement of glial phenotypes in the revised manuscript. In addition, to give a more comprehensive characterization of the glial response, transcriptome analysis of BV-2 cell line under OGD condition and with different therapeutic treatments were performed. Gene enrichment analysis of the cells based on Kyoto Encyclopedia of Genes and Genomes (KEGG) were shown in Figure 6-9, giving top 20 pathway enrichment of two selected groups for comparison. Compared with normoxia condition, the OGD-treated cells were involved in multiple biological processes of innate immunity, response to virus, cytokine signaling, suggesting a pro-inflammatory subtype. After treated by T-mPDA, BV-2 cells showed significant difference mainly in ferroptosis, which coincides with the anti-oxidative property of T-mPDA since ferroptosis is predominantly triggered by extra-mitochondrial lipid peroxidation. Moreover, this result indicated that T-mPDA may affect more on microglial survival than their pro-inflammatory phenotype. By contrast, compared to OGD alone, pathways related with retrograde endocannabinoid signaling, viral defense, innate immune response, and oxidative phosphorylation are enriched after Mino incubation. Accumulated evidence has suggested that activation of endocannabinoid signaling can suppress microglial activation (*Front. Neurol.*, 2020, 11, 87.) and microglia in anti-inflammatory states are mainly powered by oxidative phosphorylation instead of glycolysis (*Neurobiol. Dis.*, 2021, 152, 105290.). Taken together, these data suggest Mino shifted microglia to an oxidative-phosphorylation-dependent anti-inflammatory phenotype. T-mPDA-Pep-Mino incubation combined the advantages of T-mPDA (ferroptosis) and minocycline (oxidative phosphorylation, viral defense). More importantly, T-mPDA-Pep-Mino confers the additional benefits compared to T-mPDA and Mino treated separately. TNF- α signal pathway and NF- κ B signaling pathway are highly enriched after T-mPDA-Pep-Mino incubation.

Given the critical role of these pathways in pro-inflammatory cytokine production. These data imply that T-mPDA-Pep-Mino transmit BV-2 cells to an anti-inflammatory phenotype that with altered cytokine production, which is also further confirmed by quantitative real-time PCR (qPCR). Notable reduction of IL-6 and increment of IL-4 and IL-10 were recorded (Figure 10).

A heatmap of gene expression analysis determined by FPKM (Fragments Per Kilobase of transcript per Million fragments mapped) method was presented in Figure 11. As shown, the OGD treatment reduced the expression of genes in Set 1, and promoted the expression of genes in Set 3. The treatment of T-mPDA-Pep-Mino reversed the gene regulation by OGD treatment, whereas T-mPDA and Mino showed no significant impact on up- or down-regulation of OGD-treated cells gene expression. It should be noted that the genes in Set 2 that was up-regulated after T-mPDA-Pep-Mino treatment included DUSP5 (a suppressor of NF- κ B, *Sci. Rep.*, 2017, 7, 17348), SPP1 (a molecular brake suppressing the microglial inflammatory response during HIV infection, *J. Neuroinflammation*, 2020, 17, 273.), Igf1 (promotes the microglial M2 and inhibits of M1 phenotype, *Neurobiol. Aging*, 2013, 34, 1610-1620; *J. Neurochem.*, 2013, 126, 662-672.), HMOX-1 (an enzyme conferring anti-oxidative and anti-inflammatory effects, *Nat. Rev. Immunol.*, 2021, 21, 411-425), indicating the anti-inflammatory property of T-mPDA-Pep-Mino.

These results and discussion have been updated in the revised manuscript and supplementary materials. Cell sample collecting methods are also added in the section of Materials and Methods in supplementary materials. See last paragraph of Results-In vitro evaluation of MMP-2 responsive regulation of microglia polarization (Page 12).

Figure 6. Top 20 pathway enrichment determined through KEGG analysis between the groups of Normoxia and OGD.

Figure 7. Top 20 pathway enrichment determined through KEGG analysis between the groups of OGD and T-mPDA.

Figure 8. Top 20 pathway enrichment determined through KEGG analysis between the groups of OGD and Mino.

Figure 9. Top 20 pathway enrichment determined through KEGG analysis between the groups of OGD and T-mPDA-Pep-Mino.

Figure 10. Relative mRNA expression of IL-1 β , IL-6, IL-4 and IL-10 in brain tissue from the mice in different groups determined by quantitative real-time PCR ($n = 3$). ($P^* < 0.05$, $P^{**} < 0.01$ compared with the OGD group).

Figure 11. An expression heatmap of genes in BV-2 cells under different treatments.

2) The idea of the sequential/on-demand treatment is not entirely clear. It would seem that the drug will be activated as soon as it is injected after ischemic condition as there should be lots of MMP already present. It's unclear how this is different than just injecting minocycline or polydopamine directly. This should be better clarified. Perhaps some additional experiments showing MMP levels during the course of the ischemic event would be helpful to better understand the timeline for the drug activation.

Response: Thanks for kindly reminding us to clarify this point. The MMP-2 levels of MCAO mice ($n = 5$) were evaluated every 12 h after the reperfusion (0, 12, 24, 36, 48, 60, 72 h) within three days as shown below (Figure 12). From the results we can find that: 1) The MMP-2 level was elevated after the ischemia-refusion in MCAO brains as the MMP-2 concentration of Sham group was 34.8 ± 8.6 ng/mL, lower than that in any time points afterwards. 2) At each timepoint post-reperfusion, divergent MMP-2 levels from different mice in the same group were observed, which indicated the individual difference. 3) It is difficult to give a precise timepoint for minocycline administration because the MMP-2 concentration also varied significantly during different time periods. Drug

release performance from Figure 1H in the manuscript suggested that MMP-2 levels higher than 100 ng/mL could facilitate the on-demand regulation. Therefore, a responsive drug release as provided in this study is supposed to maximize the therapeutic efficacy as the drug could be activated upon the MMP-2 increment rather than the drug injection at a specific timepoint, which demonstrated the therapeutic advantages of the sequential strategy (T-mPDA-Pep-Mino) over simple combination of polydopamine and minocycline (T-mPDA+Mino). These results and discussion have been added in the revised manuscript. See 1st paragraph of Results-In vivo study of sequential therapy and brain recovery (Page 15).

Figure 12. MMP-2 concentration changes of the MCAO mice during the course of the ischemic event within three days post-reperfusion. The brain sections of euthanized MCAO mice were collected and the MMP-2 level was confirmed by ELISA assay ($n = 5$).

3) The authors suggest that although minocycline and polydopamine on their own have positive effects, the combo designed drug is more beneficial. However, is the designed drug better than just simply injecting both polydopamine and minocycline at the same time? It isn't surprising two beneficial drugs would have at least an additive effect. More details are needed to demonstrate the designed drug is better than the simple combination treatment.

Response: Thanks for your suggestions. To demonstrate the designed drug is better than the simple combination treatment, in vivo evaluation of the inflammatory cytokines by ELISA assay, glial scar observation by GFAP immunostaining, neurogenesis analysis by BrdU and DCX double staining, blood flow recovery of the MCAO mice treated by combination treatment of polydopamine and minocycline (T-mPDA+Mino) have been performed as shown below. The T-mPDA+Mino treatment showed certain therapeutic efficacy as confirmed in these evaluations. However, the sequential therapy (T-mPDA-Pep-Mino) gives more satisfactory results than the T-mPDA+Mino treatment. Compared with simple combination treatment (T-mPDA+Mino), sequential strategy induces less pro-inflammatory factors and more anti-inflammatory factors (Figure 13A); Lower level of glial scar formation (Figure 13B) and higher level of neurogenesis (Figure 13C and 13D) were observed in the group of T-mPDA-Pep-Mino. Blood flow imaging of the mice brain revealed that the T-mPDA+Mino treatment provided a better recovery of ischemic brain (Figure 13E and 13F). These profiles have been updated in the revised manuscript and the discussion on the comparison between sequential therapy and simple combination treatment was also added. See Figure 4 of the manuscript and 2nd paragraph of Discussion section (Page 17 and 21).

Figure 13. Comparative study of therapeutic efficacy of MCAO mice with different treatments including Sham, Saline, T-mPDA, Mino, T-mPDA+Mino, and T-mPDA-Pep-Mino. (A) Relative expression of IL-1 β , IL-6, IL-4 and IL-10 in brain tissue in different groups confirmed by ELISA assay ($n = 5$). (B) Glial scar evaluation by immunofluorescence GFAP staining of the ischemic penumbra for in different groups. (C) Neurogenesis evaluation by BrdU and DCX double staining and (D) corresponding analysis of the mice brains in different groups ($n = 5$). (E) Blood flow images and (F) corresponding analysis of the mice brains in different groups ($n = 3$). $P^{**} < 0.01$, $P^{***} < 0.001$, $P^{****} < 0.0001$ compared between the groups as indicated via one-way ANOVA with Tukey's multiple comparisons test. The data are presented as means \pm SEM.

Reviewers' Comments:

Reviewer #1:

Remarks to the Author:

The authors have addressed all my concerns in the revised paper. I had no more questions and recommended to accept this paper.

Reviewer #2:

Remarks to the Author:

The authors have addressed my concerns with the manuscript. I have no further comments.

Reviewer #3:

Remarks to the Author:

The authors have done a good job addressing the concerns. However, they need to remove the M1 and M2 nomenclature from the paper. These are outdated terms that confuse the field and only will detract from the study. Please see this position statement from many of the microglia experts in the field on glial nomenclature (<https://pubmed.ncbi.nlm.nih.gov/36327895/>). Microglia should be characterized by the phenotypes and characteristics they present in the specific context of the study and not lumped into M1 vs M2. By giving a more complex description, greater insight can be conveyed in the study about these specific changes.

Point-by-point Response to Reviewers' Comments

Reviewer #1

The authors have addressed all my concerns in the revised paper. I had no more questions and recommended to accept this paper.

Response: Thanks for your time re-evaluating our manuscript and your recommendation.

Reviewer #2

The authors have addressed my concerns with the manuscript. I have no further comments.

Response: We appreciate your kind help on improving our manuscript.

Reviewer #3

The authors have done a good job addressing the concerns. However, they need to remove the M1 and M2 nomenclature from the paper. These are outdated terms that confuse the field and only will detract from the study. Please see this position statement from many of the microglia experts in the field on glial nomenclature (<https://pubmed.ncbi.nlm.nih.gov/36327895/>). Microglia should be characterized by the phenotypes and characteristics they present in the specific context of the study and not lumped into M1 vs M2. By giving a more complex description, greater insight can be conveyed in the study about these specific changes.

Response: Thanks for your valuable comments. Your suggested paper (Neuron, 2022, 110(21):3458-3483) offers instructive information for microglia research. As commented, the M1 and M2 nomenclature has been removed from the manuscript and supplementary materials. Further, by following the recommendations from the Neuron paper, we revised the description of microglia in response to different treatments (as highlighted in the 1st paragraph in Introduction, 1st, 2nd, and 3rd paragraphs in Results—In vitro evaluation of MMP-2 responsive regulation of microglia polarization, and 2nd paragraph in Discussion). We hope these changes could provide an appropriate and clearer description of therapeutic effects of microglia polarization. Thanks.

Reviewers' Comments:

Reviewer #3:

Remarks to the Author:

The authors have addressed my concerns. I have no more comments.

Point-by-point Response to Reviewers' Comments

Reviewer #3

The authors have addressed my concerns. I have no more comments.

Response

Thank you for your help on improving our manuscript.